# Bootstrapping Variational Information Pursuit with Large Language and Vision Models for Interpretable Image Classification

**Aditya Chattopadhyay**[†*]    **Kwan Ho Ryan Chan**[‡*]    **René Vidal**[‡]
[†]Johns Hopkins University, USA, `achatto1@jhu.edu`
[‡]University of Pennsylvania, USA, {`ryanckh,vidalr`}`@seas.upenn.edu`

## Abstract

Variational Information Pursuit (V-IP) is an interpretable-by-design framework that makes predictions by sequentially selecting a short chain of user-defined, interpretable queries about the data that are most informative for the task. The prediction is based solely on the obtained query answers, which also serve as a faithful explanation for the prediction. Applying the framework to any task requires (i) specification of a query set, and (ii) densely annotated data with query answers to train classifiers to answer queries at test time. This limits V-IP's application to small-scale tasks where manual data annotation is feasible. In this work, we focus on image classification tasks and propose to relieve this bottleneck by leveraging pretrained language and vision models. Specifically, following recent work, we propose to use GPT, a Large Language Model, to propose semantic concepts as queries for a given classification task. To answer these queries, we propose a light-weight Concept Question-Answering network (Concept-QA) which learns to answer binary queries about semantic concepts in images. We design pseudo-labels to train our Concept-QA model using GPT and CLIP (a Vision-Language Model). Empirically, we find our Concept-QA model to be competitive with state-of-the-art VQA models in terms of answering accuracy but with an order of magnitude fewer parameters. This allows for seamless integration of Concept-QA into the V-IP framework as a fast-answering mechanism. We name this method Concept-QA+V-IP. Finally, we show on several datasets that Concept-QA+V-IP produces shorter, interpretable query chains which are more accurate than V-IP trained with CLIP-based answering systems. Code available at `https://github.com/adityac94/conceptqa_vip`.

## 1 Introduction

With the increasing complexity of modern deep network architectures, there is a growing concern over the unintelligible nature of their decision-making process (Gunning & Aha, 2019). Initial approaches to explain deep networks were post-hoc, where algorithms were designed to output scores (attributions) that reflect how "important" a particular input feature is to the network's prediction (Selvaraju et al., 2017; Lundberg & Lee, 2017; Ribeiro et al., 2016). However, these methods have been widely criticized since their explanations are often not aligned with human expectations which are more in terms of high-level semantic concepts (Koh et al., 2020; Chattopadhyay et al., 2022). Moreover, the reliability of these methods in faithfully representing the importance of different features for prediction has often been questioned (Adebayo et al., 2018; Yang & Kim, 2019).

Consequently, there is a growing need for developing deep learning methods that are interpretable by design (Koh et al., 2020; Rudin et al., 2022; Chattopadhyay et al., 2022). One line of research in this direction is based on Information Pursuit (IP) (Chattopadhyay et al., 2022; 2023). In this framework, the user first defines a set of task-specific image queries, each one having a clear interpretation to the user as a semantic question about the image. Given a new image $x^{\text{obs}}$, IP then proceeds to select queries one at a time, until the obtained answers are sufficient to make a prediction $Y$ with high

---

[*]Equal contribution

confidence. The query-answer sequence obtained upon termination serves as a faithful explanation for the prediction. This is because, IP never observes the image directly, but only through the obtained query answers, thereby guaranteeing that the prediction is made entirely on the explanation. Finally, to achieve short explanations, IP selects queries in order of information gain, that is, in each iteration the query selected has maximum mutual information with $Y$ (the prediction variable) given the history of query-answers obtained so far. An illustration of the framework is shown in Figure 1.

A central challenge in implementing IP is its reliance on mutual information, which is hard to estimate in high dimensions (Belghazi et al., 2018). To address this, a variational characterization of IP, called Variational Information Pursuit (V-IP), was proposed (Chattopadhyay et al., 2023; Covert et al., 2023). In this approach, a deep network is trained to directly learn the most informative next query from data, bypassing the need to explicitly estimate mutual information.

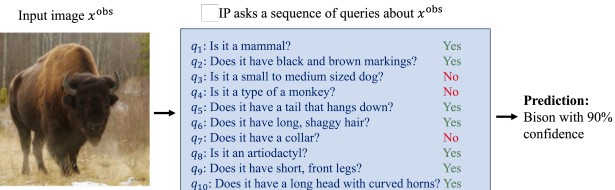

Figure 1: Illustration of the IP framework for making interpretable predictions. The task is image classification. The user-specified query set $Q$ consists of queries about the presence or absence of different semantic attributes of objects. Given an image $x^{\text{obs}}$, IP proceeds by selecting queries from $Q$ in order of information gain until a prediction can be made with high confidence.

A major limitation of V-IP is that it still requires the user to specify the query set that is relevant to the task (e.g. image classification). Moreover, this also requires datasets manually annotated with answers to the queries to allow for training classifiers which would be able to infer the answers at test time. This limits the applicability of V-IP to datasets where such annotations are available.

In this paper, we focus on image classification tasks and seek to address this limitation by employing large language models (LLMs) to specify interpretable queries for a task and vision-language models (VLMs) to answer those queries. This would allow us to scale V-IP to any image classification task without the need to manually specify query sets and their corresponding answers. To achieve this, we draw inspiration from recent work in concept-based image classification (Oikarinen et al., 2023; Yang et al., 2023; Menon & Vondrick, 2023), which shows the effectiveness of GPT (Brown et al., 2020), an LLM, at generating relevant semantic concepts that are discriminative for a given image classification task. While the semantic concepts proposed by GPT can be seamlessly integrated as interpretable queries into the V-IP framework, we still need a mechanism to answer these queries.

Prior work on concept-based image classification grounds concepts to images using CLIP (Radford et al., 2021), a VLM, which computes an $\ell_2$-normalized dot product between any given image's embedding and any given concept's text embedding as a measure of how strongly the concept is associated with the image contents, thereby answering the query whether the particular concept is present/absent in the image. Unfortunately, continuous-valued answers, as provided by CLIP, are not directly compatible with the V-IP framework since they disrupt the interpretability of the query-answer chains. As an example, consider the first query $q_1$ in Figure 1. CLIP's dot product between the image embedding of $x^{\text{obs}}$ and the text embedding for concept "mammal" is about $0.24$. It is not clear from this value if the image is of a mammal or not? This in turn makes it hard to interpret the rationale behind the selection of subsequent queries which depend on the query-answers obtained so far. This issue is not resolved by simply thresholding and binarizing the CLIP dot products since, as we show in Table 1, this results in noisy inaccurate answers when compared with the ground truth.

Our main contribution, is to show that one can effectively train a Concept Question-Answering network (Concept-QA) which produces a binary answer to the question "Is the given concept present in the image?" by utilizing soft pseudo-labels derived from both GPT and CLIP. We show via experiments that GPT complements concept-image association scores generated by CLIP since GPT has learnt from large bodies of text what concepts are salient for recognizing a particular class. We find that on several image classification datasets (via self-annotations), our Concept-QA system is competitive with contemporary pretrained VLM models used for visual question-answering tasks while being computationally much more efficient. Moreover, we empirically show that V-IP achieves much shorter query chains with higher accuracy when trained on answers provided by Concept-QA compared to answers provided by CLIP dot products thereby enhancing interpretability.

**Paper Contributions.** (1) We propose a methodology to train a concept question-answering system (called Concept-QA) which given a concept and an image, gives a binary answers as to whether the

concept is present in the image or not using pseudo-labels provided by GPT and CLIP. In particular, our method does not require any manually annotated training data grounding concepts to images. (2) We empirically validate, on multiple datasets, the performance of Concept-QA in faithfully representing the true concepts. (3) We show how Concept-QA enables us to extend V-IP's applicability to image classification datasets where manually specified query sets and answer annotations are not *a priori* available, including ImageNet (Deng et al., 2009). (4) Finally, we demonstrate that V-IP trained on answers provided by Concept-QA leads to much shorter and more interpretable query-chains than using the continuous dot products provided by CLIP as answers.

## 2 RELATED WORK

Here we briefly discuss prior work on other contemprary interpretable-by-design methods and relegate discussion of prior work on large language and vision models used in this work to Appendix §A.

**Interpretable-by-design methods without language.** In response to the criticisms levied on post-hoc interpretability methods (Adebayo et al., 2018; Kindermans et al., 2019; Slack et al., 2020; Shah et al., 2021; Rudin, 2019) many interpretable-by-design methods have been proposed. Bohle et al. (2021) propose a novel B-cos layers that incentivizes the network weights to align with task-relevant features of the input. Alvarez Melis & Jaakkola (2018) regularizes the training of deep networks such that they are locally well-approximated by linear classifiers. Li et al. (2018) proposes to learn abstract patterns from data, called prototypes, which are then linearly combined to make a prediction. These methods aim to explain predictions via input features or abstract prototypes. In sharp contrast, in this work, we seek to ground our explanations in language in terms of queries of semantic concepts, thus providing a colloquial description of the underlying models' decisions.

**Concept Bottleneck Models (CBMs).** This family of models, first introduced by Koh et al. (2020), seeks to first map a given input to a concept feature space, where every feature value corresponds to the presence or absence of the given concept. The final prediction is made by a linear classifier trained on these concept features. Our proposal in this paper to use concepts as queries for classification tasks was inspired by work on CBMs. However, V-IP and CBMs fundamentally differ in the nature of the explanations they provide for their predictions. V-IP explains the prediction by sequentially asking queries about the input in order of information gain. At each step, the selection of the next query is determined by the history of query-answers observed so far. Moreover, in each step, the user can inspect how the model's posterior over the class labels changes as more and more evidence is accumulated from each new query answer. This provides a progressive and transparent description of the model's decision-making process. CBMs on the other hand provide a static explanation by reporting the contribution (as the magnitude of the concept feature value times the weight assigned by the linear network to that feature) of every concept to the final prediction.

## 3 METHODS

### 3.1 BACKGROUND: INFORMATION PURSUIT AND VARIATIONAL INFORMATION PURSUIT

We will use capital letters to denote random variables and lower-case letters for their realizations.

**Information Pursuit (IP).** IP as an interpretable-by-design framework for making interpretable predictions for any given task was first introduced by Chattopadhyay et al. (2022). Let $X : \Omega \to \mathcal{X}$ and $Y : \Omega \to \mathcal{Y}$ denote the random variables for input data and corresponding labels/outputs, and $\Omega$ be the underlying sample space were all random variables are defined. The user first defines a query set $Q$, consisting of task-specific and interpretable queries $q : \mathcal{X} \to \mathcal{A}$, where $q(x) \in \mathcal{A}$ is the answer to the query $q \in Q$ evaluated at $x \in \mathcal{X}$. Following the specification of $Q$, the IP algorithm is described as follows: Given a data point $x^{\text{obs}} \in \mathcal{X}$, the algorithm sequentially selects queries in order of information gain, until all remaining queries are nearly uninformative. Specifically,

$$q_1 = \arg\max_{q \in Q} I(q(X); Y); \quad q_{k+1} = \arg\max_{q \in Q} I(q(X); Y \mid q_{1:k}(x^{\text{obs}})). \quad (1)$$

The symbol $I$ denotes mutual information. Here $q_{k+1}$ denotes the query selected at step $k+1$ given the history of query-answer pairs observed so far, which is defined as

$$q_{1:k}(x^{\text{obs}}) := \left\{ x' \in \mathcal{X} \mid \left( q_i, q_i(x^{\text{obs}}) \right)_{1:k} = \left( q_i, q_i(x') \right)_{1:k} \right\} \quad (2)$$

In other words, the history is the set of all realizations of $X$ that share the same answers to the first $k$ queries as $x^{\text{obs}}$. The IP algorithm terminates if all the queries in $Q$ have been asked or the remaining queries are uninformative about $Y$, conditioned on the history obtained so far. In other words, IP terminates after $L$ iterations ($L$ depends on $x^{\text{obs}}$) if either $L = |Q|$ or $\max_{q \in Q} I(q(X); Y \mid q_{1:L})) \leq \epsilon$, where $\epsilon$ is a user-defined threshold. The final prediction after termination is given by $\arg\max_{y \in \mathcal{Y}} P(Y = y \mid q_{1:L}(x^{\text{obs}}))$.

**Variational Information Pursuit (V-IP).** V-IP was introduced by Chattopadhyay et al. (2023) as an efficient approach for carrying out IP. V-IP defines a querier network $g_\eta$ (with weights $\eta$) as a function which maps arbitrary finite length query-answer chains, denoted as $s$, to a query $q \in Q$, and a predictor network $f_\theta$ (with weights $\theta$) that maps any given $s$ to a

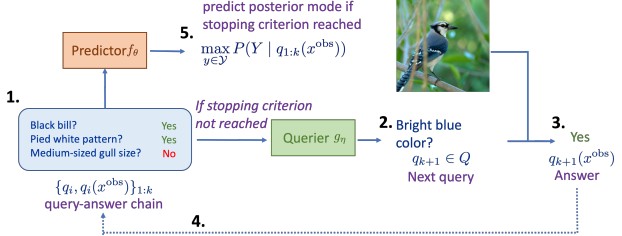

Figure 2: Overview of the V-IP algorithm

distribution over the label $Y$. Now, given any randomly selected query-answer chain $S$ and data $(X, Y)$, let the querier output some query $\tilde{q} \in Q$ and subsequently the predictor output a distribution $\tilde{P}(Y \mid S, \tilde{q}(X))$. The authors show that by minimizing the KL divergence between the true distribution $P(Y \mid X)$ and the predicted distribution $\tilde{P}(Y \mid S, \tilde{q}(X))$ in expectation over the data and random query-answer chains, one can learn to select the most informative next query given any history of query-answer pairs. More precisely, the optimal querier and predictor of the described optimization problem would be:

$$g_{\eta^*}(s) = \arg\max_{q \in Q} I(q(X); Y \mid s); \quad f_{\theta^*}(s) = P(Y \mid S = s), \tag{3}$$

where $s$ is any given query-answer chain and $P(Y \mid S = s)$ is the true posterior distribution over $Y$ given $s$. At inference time, V-IP makes predictions by using the trained querier $g_{\eta^*}$ to recursively select most informative queries (starting from the empty set at iteration 1), obtain the answer and append it to the current history of query-answer pairs observed so far. See Figure 2 for a schematic diagram of this process. The authors proposed two choices for termination, (i) Posterior-based, which terminates if the posterior is sufficiently peaked, that is, $\max_{y \in \mathcal{Y}} P(y \mid q_{1:k}(x^{\text{obs}})) \geq \gamma$, and (ii) Stability-based, which terminates if the posterior is stable for some $\kappa$ number of steps, that is, for $\kappa$ consecutive steps, the difference between the entropies of the posterior over $Y$ before and after asking a question is less than $\epsilon$ ($\gamma$, $\kappa$ & $\epsilon$ are user-defined). More details on V-IP in Appendix §E.

## 3.2 GENERATING QUERY SETS USING LANGUAGE MODELS

As discussed in §3.1, applying the V-IP framework to any task requires two ingredients, (i) specifying a set of interpretable task-relevant queries, and (ii) defining a mechanism to answer these queries for any given image at inference time. This presents a major bottleneck as curating user-defined query sets and their corresponding answers is a costly time-consuming process. In this paper, we focus on image classification tasks and seek to liberate the V-IP framework from this bottleneck by:

- Leveraging the zero-shot capabilities of GPT to propose appropriate queries for a given task.
- Training a Concept Question-Answering (Concept-QA) system using pseudo-labels provided by CLIP and GPT for answering these queries at inference time about a given image.

### 3.2.1 PROPOSING INTERPRETABLE QUERIES USING GPT

Prior work on concept-based classification methods has shown the effectiveness of GPT at generating semantic concepts that are predictive for solving image classification tasks (Oikarinen et al., 2023; Yang et al., 2023; Menon & Vondrick, 2023). These methods involve the following steps:

1. Given an image classification dataset, extract all the class labels.
2. Iteratively, for each class label, prompt GPT to specify concepts useful for describing the class. For example, if the class label is "ship", Oikarinen et al. (2023) use the prompt "List the most important features for recognizing something as a ship".

3. The above process generates a large number of candidate concepts relevant to the task. This set is then pruned to remove redundancies among the concepts. This can be done via handcrafted filtering rules (Oikarinen & Weng, 2023) or submodular optimization (Yang et al., 2023).

The assumption of these methods is that since GPT has been trained on a large corpus, it understands which semantic concepts are salient for recognizing a class. These concepts can be used as queries in the V-IP framework by converting every concept $C$ into a binary question: "Is concept $C$ present in the given image?". This ensures every query has a clear interpretation to the user since they are grounded in language. Thus, we propose to aggregate the final list of concepts (after the pruning step) and use them as queries for our query set. In this paper, for all datasets employed, we use the concepts extracted by Oikarinen et al. (2023) as our query set. Refer to Appendix Figure 7 for example extracted concepts. Having specified a procedure for automatically generating a task-relevant query set, we next discuss mechanisms to answer these queries for a given image.

### 3.2.2 ANSWERING QUERIES WITH CLIP ADVERSELY AFFECTS V-IP'S EXPLANATIONS

Prior work on concept-based classification methods proposed to use CLIP for grounding the concepts to images. This is achieved by passing the image and the concept text through CLIP's image and text encoders respectively to get their embeddings. Then, the $\ell_2$-normalized dot product between the two embeddings measures how strongly a particular concept is correlated with the image contents. Unfortunately, we will show that using CLIP's dot product as the answer to the query "Is concept $C$ present in the given image?" hampers the interpretability of V-IP's reasoning process.

Recall that V-IP's explanation of a prediction is the sequence of query-answer pairs. The choice of every query depends only on the prior history of query-answers. Interpreting this reasoning process crucially depends on the interpretation of this history. We claim that this history (equation 2) is *interpretable* to the user only if given any image, the user can ascertain if the image belongs to the history. We motivate this definition using Figure 1. The first question asked is whether the image is of a mammal, and the answer is "Yes". The user then knows that given any image sampled from the data distribution, if it contains a mammal, it will belong to the model's history. Continuing like this, we see the model ask several other questions about different characteristics of mammals like hair texture, tail shape, etc. before it concludes it is a bison. Since, at every step, the history is interpretable, V-IP's

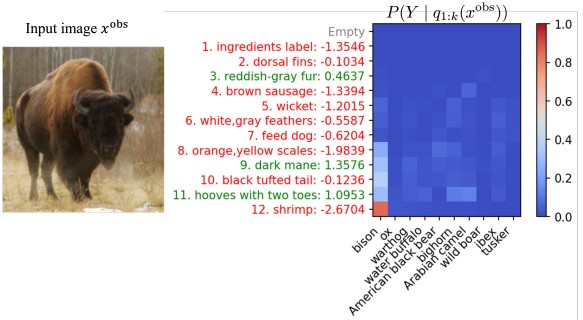

Figure 3: V-IP's explanation for an image from the Imagenet validation set using CLIP's dot products as answers. The input image, $x^{\text{obs}}$, is shown on the left and the query-answer chain is shown on the right along with the evolution of the model's posterior distribution. Every row corresponds to the posterior distribution over the classes (shown in the x-axis) given the history of query-answer pairs observed so far. The y-axis indicates the query asked at that iteration along with CLIP's dot product answer. For brevity, we only mention the concept, instead of the corresponding query "Is the given concept present in the image?" (best viewed in colour).

reasoning up to the prediction of "bison" is transparent. However, when the answers are continuous-valued dot products, as supplied by CLIP, the history is no longer interpretable (according to our definition). We elaborate on this next with an example.

We train V-IP on the Imagenet dataset using the query set described in §3.2.1 and CLIP's dot products (standardized by subtracting and dividing by the mean and std computed over image-concept pair from the entire dataset) to answer the queries. In Figure 3, we depict our trained V-IP's explanation for an image from the Imagenet validation set. Positive and negative dot products are colour coded as green and red respectively. The first query, "Is the concept ingredients label present in the image?", gets an answer of $-1.35$. According to our definition, the history after obtaining this answer, $q_1(x^{\text{obs}})$, is not interpretable to the user since we do not know what images would give the same dot product of $-1.35$ when asking the same question. The choice of the second query about dorsal fins as well as the remaining queries, are hard to interpret. Moreover, the final query about "shrimp" makes no sense given the posterior after 11 queries was peaked on "bison", "bighorn", and "Arabian camel". We argue this is because histories in this context are not interpretable.

### 3.2.3 OUR PROPOSAL: CONCEPT QUESTION-ANSWERING NETWORK

One possible solution to solving this interpretability conundrum is to make the query answers discrete, with every answer having a well-defined interpretation to the user. However, in the absence of data annotated with query answers, how can one learn a classifier that would "accurately" predict the answer for a given image at test time? To answer this, we rely on the zero-shot capabilities of GPT and CLIP.

A first attempt at this might be to simply threshold the standardized CLIP dot products and interpret a positive value as the concept being present in the image. However, as we will empirically show in §4, such a method results in inaccurate answers (compared with the ground truth) and makes them difficult to interpret. We thus propose to complement the knowledge CLIP has about image-concept association with GPT's knowledge about different concepts that are important for recognizing a particular class. In particular, we train a Concept Question-Answering network (Concept-QA) using pseudo-labels (which are our target answers for training Concept-QA) generated from GPT and CLIP. We first describe the intuition behind these pseudo-answers and then explicitly define them.

In image classification tasks, the class is often the focal point of the image. This means that *a priori* before inspecting the image, if we know the class then we can be pretty confident that some concepts will not be present in the image. For example, if we know the image is of class "Tiger", we know that it is unlikely the concept "a garment" will be present in the image. However, answering a query about a concept like "white" requires inspecting the image contents since some tigers can have white fur, while others are predominantly orange. We formalize this intuition using GPT and CLIP. Specifically, for a given image classification dataset, we take every image-label pair in the corresponding train set and construct image-concept-pseudo-label triplets for all concepts in our query set in the following manner:

1. For every image-label pair and concept in our query set, we ask GPT if the concept is salient for recognizing the class label.
2. If GPT responds "No", we take the pseudo-label for that concept-image pair as "No".
3. If GPT responds "Yes/Depends", we use CLIP's dot product between the concept's text embedding and the image's embedding to disambiguate whether the concept is present in the image. Since CLIP's dot product is a measure of how strongly the concept is aligned with the image content, we use it (after suitable normalization which we will describe next) to model the probability that the concept is present in the image.

Moving forward, we would use the value 1 (0) interchangeably with "Yes" ("No") to talk about the answer/prediction for a particular concept query. Now, given image-label pair $(x, y) \in \mathcal{X} \times \mathcal{Y}$ and a binary concept $C$ which corresponds to a query $q_C \in Q$, we define the following terms,

$$P_{\text{GPT}}(C = 1 \mid Y = y) = 1 \quad \text{if GPT responds "Yes" to the above mentioned prompt}$$
$$= 0 \quad \text{otherwise} \tag{4}$$
$$P_{\text{CLIP}}(C = 1 \mid X = x) = \Phi(\mathcal{I}(x)^T \mathcal{T}(c)),$$

where $\mathcal{I}(x)$ and $\mathcal{T}(c)$ correspond to the $\ell_2$-normalized image and concept's text embedding respectively using CLIP's encoders. $\Phi(.)$ is a per image min-max normalization function which ensures that the concept with the highest dot product with the $x$ is given a probability score of 1, and the concept with the smallest dot product with $x$ is given a probability score of 0. The two probabilities from equation 4 are then combined to give the pseudo-label for the pair $(x, y)$ and concept $C$ as,

$$\hat{P}(C = 1 \mid X = x, Y = y) = P_{\text{GPT}}(C = 1 \mid Y = y)P_{\text{CLIP}}(C = 1 \mid X = x) \tag{5}$$
$$\hat{P}(C = 0 \mid X = x, Y = y) = P_{\text{GPT}}(C = 0 \mid Y = y) + P_{\text{GPT}}(C = 1 \mid Y = y)P_{\text{CLIP}}(C = 0 \mid X = x).$$

Finally, we propose to train a deep network, called Concept-QA, which takes CLIP's embeddings of the image and the concept's text and makes a binary yes/no prediction to the query "Is the given concept present in the given image?". We use the generated pseudo-labels, $\hat{P}$, to train our Concept-QA network using the following loss function for a given concept C (which is reminiscent of the popular cross-entropy loss when the target labels are one-hot),

$$\text{Loss(C)} = -\mathbb{E}_{X,Y}\Big[ \sum_{c \in \{0,1\}} \hat{P}(C = c \mid X, Y) \log P_w(C = c \mid X)\Big], \tag{6}$$

where $w$ defines the Concept-QA model weights. Notice our model predicts whether the concept $C$ is present in the image purely based on the image contents (no $Y$ information is provided), as embodied in the notation $P_w(C = c \mid X)$. The optimization problem is defined as, $\min_w \sum_{C \in Q} \text{Loss}(C)$, where the summation is over all concepts (or their corresponding queries) in our given query set $Q$. Figure 4 gives an overview of Concept-QA training. More training details in Appendix §D.

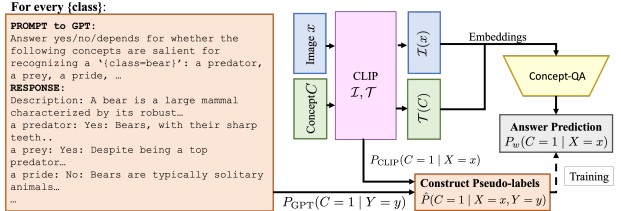

Figure 4: Overall pipeline for training and performing inference on the Concept-QA system. Details about the GPT and CLIP prompts used in our experiments can be found in Appendix §D.

After training, for a given image $x^{\text{obs}}$ and any concept $C$, we take the corresponding query's answer $q_C(x^{\text{obs}}) = 1$ if $P_w(C = 1 \mid X = x^{\text{obs}}) > \tau$ and 0 otherwise. The constant $\tau$ is a user-defined parameter which determines how confident we want the Concept-QA to be in its prediction of 1 before accepting it as the binary answer. We use $\tau = 0.4$ for all datasets considered in this paper except CIFAR-$\{10, 100\}$ for which we consider $\tau = 0.5$.

## 4 EXPERIMENTS

In this section, we empirically evaluate the efficacy of our proposed Concept-QA model in extending the application of V-IP to image classification tasks where manually specified query sets and answer annotations are not available. We do so in two complementary directions.

1. **Interpretability of query sets.** In prior work (Chattopadhyay et al., 2022; 2023) the query sets were manually specified and every query had a clear interpretation to the user. However, in this work, we train our Concept-QA model using pseudo-labels to answer different queries. Thus, the interpretability of our queries depend on how faithfully Concept-QA reflects the ground truth. For example, if for some query "Is concept 'stripes' present in the image?", Concept-QA erroneously predicts "1" (corresponding to a "Yes") when the image has the concept "checkerboard pattern" in it (but no stripes), then the answer no longer has a clear interpretation to the user.

2. **Description length and accuracy of V-IP explanations.** Consequently, given an interpretable query set, V-IP makes predictions by asking a sequence of query-answer pairs about a given input. This sequence serves as an explanation for the prediction. In accordance with Occam's razor, Chattopadhyay et al. (2022) advocate that shorter explanations are preferred over longer ones if both achieve the same prediction accuracy. Thus, it is important to have query sets that result in both short and accurate explanations.

Finally, we provide qualitative examples of explanations provided by V-IP using Concept-QA as an answering mechanism and contrast it with the uninterpretable query-answer chains obtained when using CLIP dot products as the answering mechanism.

We carry out all our experiments on 5 popular image classification datasets; ImageNet (Deng et al., 2009), CUB-200 (Wah et al., 2011), Places365 (Zhou et al., 2017), CIFAR-$\{10,100\}$ (Krizhevsky et al., 2009). In all our experiments involving CLIP, we used the `ViT-B/16` backbone, and for GPT we used the `gpt-3.5-turbo-0613` version, except for CUB-200 where we found GPT-3.5 ineffective due to the domain-specific nature of the dataset and opted to use `gpt-4` instead. For every dataset, the query set employed is the set of concepts extracted by Oikarinen et al. (2023).

**Interpretability of query sets.** We evaluate how faithful Concept-QA is in representing the true answers and compare its performance with the following baselines: (i) CLIP-Bin$_{\text{std}}$, which uses the binarized CLIP dot products (standardized as explained in §3.2.2) as answers with a positive/negative value taken as 1/0. (ii)

Table 1: Results for evaluating the faithfulness of different answering mechanisms to the true answers. Acc. and F1 refer to accuracy and F1-score metrics respectively.

| Model | ImageNet | | Places365 | | CUB-200 | | CIFAR-10 | | CIFAR-100 | |
|---|---|---|---|---|---|---|---|---|---|---|
| | Acc. | F$_1$ | Acc. | F$_1$ | Acc. | F$_1$ | Acc. | F$_1$ | Acc. | F$_1$ |
| Concept-QA (Ours) | **0.87** | **0.56** | 0.83 | 0.45 | **0.80** | **0.54** | 0.80 | **0.62** | 0.80 | 0.38 |
| CLIP-Bin$_{\text{std}}$ | 0.55 | 0.39 | 0.58 | 0.42 | 0.56 | 0.48 | 0.58 | 0.47 | 0.51 | 0.21 |
| CLIP-Bin$_{\text{norm}}$ | 0.50 | 0.27 | 0.49 | 0.26 | 0.56 | 0.45 | 0.66 | 0.53 | 0.54 | 0.24 |
| BLIP2 ViT-g OPT$_{2.7B}$ | 0.55 | 0.31 | 0.76 | 0.18 | 0.53 | 0.35 | 0.73 | 0.13 | 0.86 | 0.07 |
| BLIP2 ViT-g FlanT5$_{XL}$ | 0.86 | **0.56** | **0.87** | **0.62** | 0.70 | 0.40 | **0.83** | 0.59 | **0.87** | **0.41** |

CLIP-Bin$_{norm}$, which obtains answers by thresholding $P_{CLIP}(C=1 \mid X=x)$ (equation 4) at $0.5$. (iii) BLIP2 ViT-g FlanT5$_{XL}$ & BLIP2 ViT-g OPT$_{2.7B}$, which are state-of-the-art pre-trained VLMs that are effective in zero-shot Visual Question-Answering (VQA) tasks (Li et al., 2023).[1] Since the datasets we consider do not come with annotated query answers we randomly sample $2.5K$ image-concept pairs from each dataset and self-annotate them with the true answers. Details of the sampling and annotation process can be found in Appendix §C. For each dataset, we report the accuracy and F1-score and report the results in Table 1.

We observe that Concept-QA outperforms CLIP-Bin$_{std}$ and CLIP-Bin$_{norm}$ on all 5 datasets on both metrics, supporting our claim that simply binarizing CLIP dot products based on a threshold will lead to noisy answers and integrating GPT with CLIP to provide the pseudo-labels enhances the quality of the supervision signal for training the answering model. Furthermore, compared to the BLIP2 FlanT5 model, Concept-QA is competitive on the CIFAR datasets and ImageNet. Our model is considerably better on CUB-200, a fine-grained bird species identification dataset which is too specific for the general-purpose VQA model to have good zero-shot performance. Finally, BLIP2 FlanT5 performs better on the Places365 dataset which is a scene classification dataset. We hypothesize that this is due to the inability of CLIP's image encoder at capturing essential image semantics for this dataset. Table 2 in the Appendix corroborates this, which shows that a linear probe trained on CLIP's image embedding achieves a test accuracy of only $55\%$ which is much lower than the accuracy achieved on all the other datasets.

While being competitive with BLIP2 on most datasets, Concept-QA is much more computationally efficient than these large VLMs which have billions of parameters. Concretely, ConceptQA is a lightweight MLP with 70K parameters which operates on top of CLIP resulting in a total of 150M parameters. The competitive VLM, BLIP2 FlanT5, on the other hand has about 4.1B parameters. Consequently, our model takes about 0.04 seconds per image per query, while the VLM takes about 1.52 seconds for the same. This makes the latter prohibitive for utilization within the V-IP framework for training and inference. For example, carrying out one epoch of V-IP training on Imagenet using BLIP2 Flan-T5 as the answering mechanism would take about $4,128$ GPU hours (evaluated on NVIDIA RTX A5000 GPU by averaging the time taken to process 100 batches of image-query pairs and multiplying by the size of the dataset.) which is infeasible. Comparatively, V-IP with Concept-QA takes just about $0.5$ GPU hours to finish one epoch! Similar computational challenges also plague the use of VLMs to directly provide the pseudo-labels for training the Concept-QA model in place of our proposed pseudo-labels (equation 5). Refer Appendix §F for a discussion on this.

**Description length and accuracy of V-IP explanations.** Next, we analyze the effectiveness of V-IP in obtaining short and accurate explanations using our query sets and compare different answering mechanisms. Following are the models considered: (i) V-IP trained with Concept-QA, Concept-QA+V-IP; (ii) V-IP trained with CLIP continuous-valued dot products (standardized), CLIP+V-IP; (iii) V-IP trained with CLIP-Bin$_{std}$, CLIP-Bin$_{std}$+V-IP; (iv) V-IP trained with CLIP-Bin$_{norm}$, CLIP-Bin$_{norm}$+V-IP; and (v) V-IP trained with BLIP2-FlanT5; BLIP2-FlanT5+V-IP.

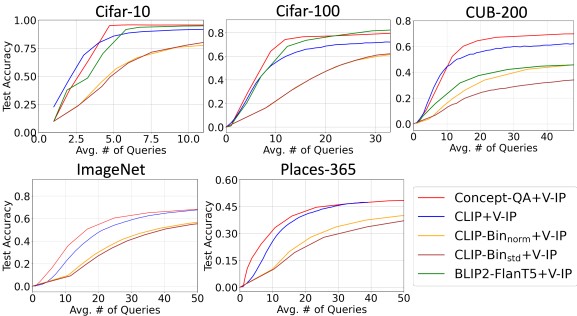

Figure 5: Trade-off between test accuracy vs. avg. # of queries (explanation length). View in colour.

For training and architecture details for these models refer to Appendix §E. We report our results in Figure 5. For each model, the curve was generated by selecting the stopping criterion that yielded the best results: the stability-based criterion (see Section §3.1) for Concept-QA+V-IP and BLIP2-FlanT5+V-IP, and the posterior-based criterion for the others.

We see in Figure 6 that Concept-QA+V-IP outperforms CLIP+V-IP (along with its binarized CLIP-Bin variants) on all datasets in terms of the average number of queries needed to reach a desired level of accuracy (except on CIFAR-10 for a small "low test accuracy, low avg. number of queries" regime). As articulated previously, state-of-the-art VLMs are computationally intensive and thus,

---

[1]For both these VLMs we use the prompt "Question: Is `concept` present in the image (Yes/No)? Answer:", where `concept` is replaced by the name of the concept we are interested in asking the VLM about.

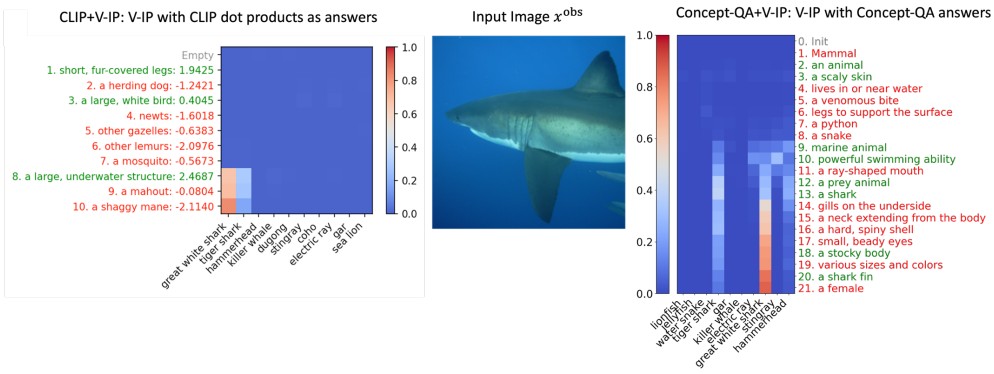

Figure 6: Example run of the V-IP algorithm on an example image from the validation set of ImageNet. Column 1 shows CLIP+V-IP trajectories, while Column 3 Concept-QA+V-IP trajectories. Refer to the caption of Figure 3 for a description of the heatmaps. The x-axis of every heatmap shows the top-10 classes according to the posterior after termination. For queries/concepts, a green (red) colour denotes a positive (negative) dot product in col 1 and a "Yes"("No") answer in col 3. View in colour. See Appendix §G for more examples.

inefficient to use for answering queries in V-IP training for large datasets like Imagenet. As a result, we could only compare with BLIP2-FlanT5 on the three relatively smaller datasets. Our result indicates that Concept-QA+V-IP outperforms BLIP2-FlanT5+V-IP on CIFAR-10 and CUB-200 (the latter by a huge margin). On CIFAR-100, our model has a better trade-off for short explanation lengths but eventually the BLIP2-FlanT5+V-IP gets better results. Interestingly, these observations correlate well with the results in Table 1 where the answers supplied by Concept-QA are more faithful to ground truth than the VLM on CUB-200, competitive on Cifar-10 and slightly worse on Cifar-100. Finally, since interpretability can be seen as a constraint on learning, we discuss in Appendix §B.1 the gap between V-IP's performance using query sets to that of a black-box model.

**Interpretability of Predictions by V-IP.** Having quantitatively established the efficacy of our proposed Concept-QA with V-IP, we now present qualitative examples illustrating example runs of V-IP (See Figure 6 and more examples in Appendix §G). Our examples show a sharp contrast in interpretable explanations when answers are supplied by Concept-QA vs. uninterpretable explanations generated when answers are supplied by CLIP continuous-valued dot products.

Observe, in Figure 6 Concept-QA+V-IP starts by asking if the image is of a mammal. This first query is independent of the image contents (see equation 1). Upon receiving a "No" answer, it proceeds to ask if it is an animal and if it has scaly skin. Since both these answers were "Yes", the model knows the image is of an animal which has scales. The next query asks whether the animal lives near water, for which the Concept-QA model incorrectly says "No". This leads Concept-QA+V-IP down the wrong branch where it asks questions to see if the animal is a snake (which does not necessarily live near water bodies). After repeated "No" responses, Concept-QA+V-IP circles back and asks if the animal is a marine animal, which gets a positive response from Concept-QA. Finally, after query 13, Concept-QA+V-IP knows the image is of a shark and we see the posterior distribution concentrate all its mass on three different species of sharks. Subsequently, the class "hammerhead" is eliminated from consideration after asking whether the shark has a hard, spiny shell; a neck extending from the body and small, beady eyes - all characteristics of a hammerhead shark. Finally, Concept-QA+V-IP disambiguates between the great white and tiger shark by asking whether the shark has a "stocky body", which is a characteristic feature of great white sharks compared with tiger sharks which have relatively sleeker bodies. Such rich transparent elucidation of the decision-making process in V-IP is lost when the answers are taken as dot products. This is clear by observing the explanation provided by CLIP+V-IP for the same image. Since the answers are now dot products, the queries selected by V-IP are no longer the same. For example, after 8 queries, CLIP+V-IP has all its posterior mass on great white and tiger sharks. However, the next two queries are about a mahout and a shaggy mane, neither of which are related to the two shark species in consideration. However, this information is used by CLIP+V-IP to conclude the image is of a great shark with more than $80\%$ confidence. We conclude this section by directing the reader to §H (in the Appendix) for a discussion on limitations of our current approach and potential future directions.

## ACKNOWLEDGMENTS

This research was supported by the Army Research Office under the Multidisciplinary University Research Initiative contract W911NF-17-1-0304, the NSF grant 2031985 and by Simons Foundation Mathematical and Scientific Foundations of Deep Learning (MoDL) grant 135615. Moreover, the authors acknowledge support from the National Science Foundation Graduate Research Fellowship Program under Grant No. DGE2139757. Any opinions, findings, and conclusions or recommendations expressed in this material are those of the author(s) and do not necessarily reflect the views of the National Science Foundation.

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

APPENDIX

## A    RELATED WORK ON LARGE LANGUAGE AND VISION MODELS

OpenAI's GPT-3 (Brown et al., 2020) is an LLM trained on large amounts of unlabeled text in a self-supervised manner. Recent benchmarks have demonstrated that LLMs can perform complex human tasks, such as passing the Bar Exam (Bommarito II & Katz, 2022) and show signs of general understanding (Bubeck et al., 2023). Adding other data modalities, CLIP (Radford et al., 2021) is a VLM pre-trained on millions of image-text pairs. CLIP has been a building block for many recent VLMs that achieve state-of-the-art zero-shot performance in image classification (Menon & Vondrick, 2023), Visual Question Answering (Eslami et al., 2021; Parelli et al., 2023; Li et al., 2023), and Video Question Answering (Ye et al., 2023). Also known as Foundational Models (Bommasani et al., 2021), LLMs and VLMs have demonstrated strong capabilities to learn complex concepts, which motivates our proposal to use GPT and CLIP to propose query sets for V-IP.

## B    EXTENDED RESULTS

### B.1    ACCURACY OF CONCEPT-QA+V-IP VS. BLACK-BOX MODELS

We compare the test accuracy obtained by of Concept-QA+V-IP vs. black-box non interpretable models in Table 2. Recall that V-IP obtains query answer chains by sequentially selecting the most informative features until a stopping criteria is reached. For every dataset, we use the stability criterion with parameter $\gamma$ and $\kappa$: the stability criterion is reached when the difference between two consecutive posterior entropies is below $\gamma$ for $\kappa$ consecutive iterations. In practice, the user chooses the parameters for the stopping criterion by balancing the trade-off between the number of queries used to explain the prediction and the accuracy of the prediction: a longer query chain can potentially achieve a better accuracy but at the cost of interpretability due to longer query-answer chains. On the extreme case, we evaluated the test accuracy of Concept-QA+V-IP with all available query answers $Q(X)$ are used, which corresponds to the upper limit to how well Concept-QA+V-IP can perform with respect to using a variable number of queries. Next, we compare Concept-QA+V-IP's accuracy with vision-only baselines to ascertain how much accuracy is sacrificed due to the constraint of interpretability (through query sets). For a fair comparison, we compare with CLIP's ViT-B/16 image encoder since the Concept-QA network takes as input the image embeddings produced by this encoder. We evaluate the test accuracy of a using Linear Probe on CLIP's ViT-B/16 (which solves a logistic regression problem using image embeddings of images as input features) since this was the metric reported in the original CLIP paper (Radford et al., 2021). Moreover, we also compare with the accuracy the ViT-B/16 vision transformer achieves on each of these datasets (supervised training baseline). Table 2 shows that for all datasets Concept-QA+V-IP is able to achieve a test accuracy that is close to vision-only baselines, with the exception of CUB-200 where an appreciable gap remains (about 20%).

Table 2: Test Performance of Concept-QA+V-IP. The second column shows the sizes of the query set. Explanation Length is the number of queries needed on average over all test samples to reach the stopping criterion. Accuracy of Concept-QA+V-IP's prediction after the stopping criteria is reached is given in column five. In column six we report the accuracy of Concept-QA+V-IP using all available query answers $Q(X)$. Accuracy on pretrained CLIP ViT-B/16 using linear probe is shown in column seven. Finally, we report the supervised classification accuracy of the ViT-B/16 vision transformer in the last column.

| Dataset | $|Q|$ | Stopping Criterion $(\gamma, \kappa)$ | Explanation Length | V-IP Acc. Given $(\gamma, \kappa)$ | V-IP Acc. Given $Q(X)$ | CLIP ViT-B/16 Linear Probe | ViT-B/16 |
|---|---|---|---|---|---|---|---|
| CUB-200 | 208 | Stability (0.127, 4) | 24.7 | 67.0 | 70.8 | 80.0 | 90.6 (Demidov et al., 2023) |
| CIFAR-10 | 128 | Stability (0.127, 4) | 5.5 | 95.6 | 95.7 | 96.2 | 99.0 (Dosovitskiy et al., 2020) |
| CIFAR-100 | 824 | Stability (0.127, 4) | 21.3 | 77.3 | 80.2 | 83.1 | 91.9 (Dosovitskiy et al., 2020) |
| ImageNet | 4523 | Stability (0.127, 10) | 49.6 | 68.2 | 73.8 | 80.2 | 84.2 (Dosovitskiy et al., 2020) |
| Places365 | 2207 | Stability (0.127, 10) | 32.1 | 46.5 | 51.5 | 55.1 | 58.2 (Singh et al., 2022) |

### B.2    EXAMPLES OF QUERIES $q_C$

See Figure 7. Recall, for every concept $C$, the corresponding query $q_C$ is "Is concept C present in the image?"

**CUB-200**

1. a blue sheen on the wings
2. all-dark plumage
3. a thin, black bill
4. black cap and white "eyeline"
5. bright golden-yellow plumage
6. fast, erratic flight patterns
7. orange and black wings
8. pale gray or white color
9. red legs
10. yellow tips on the wings
11. black and white plumage
12. a small, black bird

**CIFAR-10**

1. a barn
2. a beetle
3. a driver
4. a long neck
5. a seatbelt
6. four legs
7. taillights
8. A living thing
9. A cargo
10. a wet nose
11. Feathers
12. a mosquito

**CIFAR-100**

1. a alarm clock
2. a bandit
3. a bathing suit
4. a deer
5. a keypad
6. a jungle
7. a large number of trees
8. a large, rounded bottom
9. a lobster trap
10. a long stem with thorns
11. a mirror
12. a phone book

**ImageNet**

1. a small, round disk at one end
2. a hitch for towing
3. a sleek, metallic exterior spacecraft
4. a curved or V-shaped bottom
5. a dress
6. a pump
7. a buffalo
8. a erect ears
9. protective gear
10. a doctor
11. a large, rotund body
12. a long, snake-like shape

**Places365**

1. paved with asphalt or concrete
2. a large, cabin-like structure
3. white or light-colored walls
4. Hair
5. a band
6. a tennis court
7. a brightly lit interior
8. pipes
9. has a crater at the top
10. a gas cap
11. a sign that says "no dumping"
12. a family of penguins

Figure 7: Sampled queries from the query set generated by GPT. Query generation process is discussed in Appendix D.

## C    DETAILS OF USER-ANNOTATED QUERY ANSWERS

**Survey Creation.** Each dataset is given a query set $Q$ and has 1000 images randomly selected from the test set. For each VQA system, all queries $q_C \in Q$ are evaluated on the images sampled. Given the set of all query answers, we sample 250 positive image-query pairs where the VQA system answers "yes" ($q_C(x) = 1$) and 250 negative image-query pairs where the answer is "no" ($q_C(x) = 0$). Hence, a given dataset and VQA system should have 500 image-query pairs with equal number of positive and negative query answers pairs. For evaluation, each VQA system is evaluated on the 2,000 image-query pairs collectively gathered from the four different VQA systems. Since the performance of VQA systems may vary query to query, our sampling processing ensures that the evaluation set does not favor any of the VQA system by oversampling with positive/negative query-answer pairs from any single system.

**Survey format.** Each evaluation requires a sampled image $x$ and a query $q_C$. The evaluator (the person performing the evaluation) is presented a picture of the plotted image and question "Is the concept {concept} present in the image?" as title, with {concept} replaced by the concept being evaluated. The evaluator is asked to answer only in yes or no. Two examples for evaluating ImageNet are presented in Figure 8.

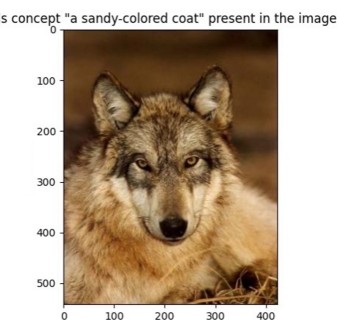
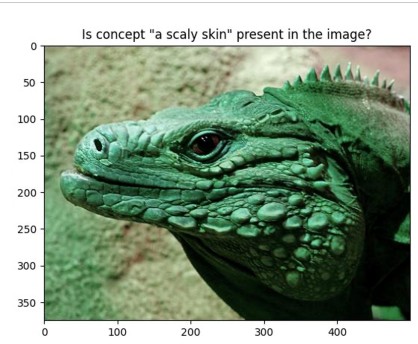

Figure 8: Two examples of pictures presented to the evaluator to evaluate the presence of a concept $C$ in a given image $x$.

**Providing Labels.** Combining the surveys for each dataset and for each model, there are a total of 10,000 individual evaluations of to answer. Every evaluation is completed by the authors of this work over the span of two weeks. On average, each person spends about 20 mintues to perform 500 image-concept evaluations.

## D   IMPLEMENTATION: CONCEPT-QA

**Query Set Generation.** To obtain a task-relevant and interpretable query set, we follow the practice Oikarinen et al. (2023) to prompt GPT. We briefly reiterate the process of prompting GPT and filtering concepts below:

For a given dataset, we first query GPT-3 with the following prompts:

- List the most important features for recognizing something as a {class}
- List the things most commonly seen around a {class}
- Give superclasses for the word {class}

where {class} is the name of a class. We query GPT with this prompt layout for every class. Then, a number of filters are applied the the set of concepts generated to improve the interpretability of the concepts. For instance, concepts with more than 30 characters are removed, and concepts that are too similar to class names are also removed. For filtering details, please refer directly to Oikarinen et al. (2023). The resulting sizes for the set of concepts used in our following experiments can be found in Table 2 (second column to the left).

**pseudo-labels from GPT.** To obtain pseudo-labels from GPT, we use OpenAI's API to prompt GPT with the following lines of text:

```
Answer yes/no/depends for whether the following concepts are salient
for recognizing a '{class}': {concept1}, {concept2}, ..., {conceptN}.
Output format: <concept>: <answer>: <explanation>. Answer as a list.
```

where {class} is the name of the class, {concept1}, {concept2}, ..., {conceptN} corresponds to the list of concepts generated, separated by commas, assuming there are a total of N concepts. We repeat this prompting process for every possible class of a given dataset. The number of concepts N varies for each dataset.

**Architecture.** The network architecture of the Concept-QA is the same for every dataset and a diagram of its architecture is shown in Figure 9. Inputs to the network are CLIP image embedding $\mathcal{I}(x)$ of a given image $x$ and text embeddings $\mathcal{T}(C)$ for a given concept $C$. The assume that $\mathcal{T}(x)$ and $\mathcal{I}(C)$ are $\ell_2$-normalized. Output of the Concept-QA is a scalar-valued probability, indicating whether the concept $C$ is present in the image $x$.

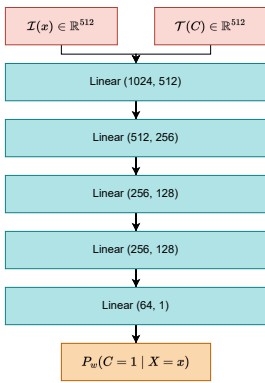

Figure 9: Architecture of Concept-QA. ReLU nonlinearity followed by a BatchNorm are used between each Linear layer.

**Optimization.** For each dataset, Concept-QA is optimized using the Binary Cross Entropy objective. For CIFAR-10, the network is optimized using Stochastic Gradient Descent (SGD) with learning rate 0.01, momentum 0.9 and weight decay $5 \cdot 10^{-4}$. For CIFAR-100, the network is also optimized using Stochastic Gradient Descent (SGD), but with learning rate 0.1, momentum 0.9 with `Nesterov=True`, weight decay $5 \cdot 10^{-4}$. For CUB, the optimizer is the same as CIFAR-10, except momentum 0.9 with `Nesterov=False`. For ImageNet and Places365, the network is optimized with learning rate $10^{-4}$, weight decay 0, and `amsgrad=True`. For learning rate schedulers, each dataset uses a Cosine Annealing Learning Rate scheduler with `T_max=200`.

## E  IMPLEMENTATION: V-IP

We specify the details of the the query answers, architectures and optimization methods used for training Concept-QA+V-IP in Appendix E.4 and training CLIP+V-IP in Appendix E.1. Meanwhile, the V-IP objective and the way we represent histories are stated below.

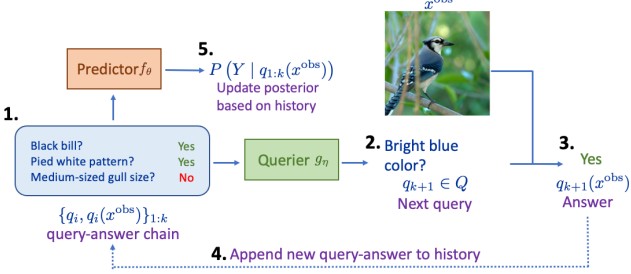

Figure 10: Overview of the V-IP algorithm.

**V-IP Objective.** V-IP was introduced by Chattopadhyay et al. (2023) as a variational approach to IP which is very efficient in practice. V-IP defines a predictor $f : \mathcal{S} \to \mathcal{P}(Y)$ and a querier $g : \mathcal{S} \to Q$ that map a query-answer chain $s \in \mathcal{S}$ of any finite length to a distribution over $Y$ ($\mathcal{P}(Y)$ denotes the set of all possible distributions on $Y$) and query $q \in Q$, respectively. Parameterized by deep networks $\theta$ and $\eta$, $f_\theta$ and $g_\eta$ are trained by sampling random query-answer chains and optimizing

the following V-IP objective:

$$\min_{\theta,\eta} \quad \mathbb{E}_{X,S}[D_{\mathrm{KL}}(P(Y \mid X)\|P_\theta(Y \mid q_\eta(X), S)] \qquad (7)$$

$$\text{where} \quad q_\eta := g_\eta(S)$$

$$P_\theta(Y \mid q_\eta(X), S) := f_\theta(\{q_\eta, q_\eta(X) \cup S\}),$$

where $P(Y = y \mid X = x)$ is the true posterior over $Y$ given input $X$. Throughout this paper, conditioning on event $S = s$ should be understood as the set of all datapoints in $\mathcal{X}$ that share the same answers to queries as in $s$. Both Concept-QA+V-IP and CLIP+V-IP are learned by solving for $\theta$ and $\eta$ in the optimization problem stated in equation 7.

**Representing and Updating the History $S$.** In V-IP framework, the input to predictor $f_\theta$ and querier $g_\eta$ are both histories $S$. Similar to Chattopadhyay et al. (2023), the history for a sample $x$, $S(x)$, is represented as the product of $|Q|$-dimensional feature vector for all query answers $Q(x)$ and a binary mask $M$, i.e. $Q(x) \odot M$, where $\odot$ here represents point-wise multiplication, also known as the Hadamard product. The $i$-th dimension corresponds to the query answer $q_i(x)$. We set $M_i$ to 0 if the $q_i(x) \notin S$ and 1 if $q_i(x) \in S$. Hence, a history with all query answers has $M$ equal to a vector of all ones, and an empty history is represented by a mask of all zeros.

Suppose we have a history for sample $x$ of size $k$, $S_k(x) = Q(x) \cdot M_k$. To update the history $S_k$ with an additional query from the output of the querier $g_\eta(S_k)$, we simply update the mask and the history by:

$$M_{k+1} \leftarrow M_k + g_\eta(S_k) \quad \text{and} \quad S_{k+1} \leftarrow S_k + Q(x) \odot g_\eta(S_k)$$

In other words, the new query is mathematically represented as a one-hot vector with dimension $|Q|$, hence updating the representation of history is equivalent to updating the binary mask $M$ by setting $i$-th position to 1, indicating that the $q_i(x)$ is in the updated history $S_{k+1}(x)$.

## E.1 TRAINING CLIP+V-IP

**Query Answers $Q(X)$.** Query answers are computed using CLIP dot-products. Again, we denote $\mathcal{I}(\cdot)$ and $\mathcal{T}(\cdot)$ as the image and text encoder from CLIP. For any task and dataset, we obtain the image embeddings of all images $\mathcal{I}(x) \; \forall x \in \mathcal{X}$ and $\mathcal{T}(q) \; \forall q \in Q$. Each embedding is assumed to be $\ell_2$-normalized, then the query answer $q(x) = \mathcal{I}(x) \cdot \mathcal{T}(q)$. We further Z-score standardize each query answer by subtracting the mean and dividing the standard deviation of query answers from the training set:

$$q(x) \leftarrow \frac{q(x) - \hat{\mu}}{\hat{\sigma}}, \quad \hat{\mu} = \frac{1}{|\mathcal{X}||Q|} \sum_{x \in \mathcal{X}} \sum_{q \in Q} q(x), \quad \hat{\sigma} = \sqrt{\sum_{x \in \mathcal{X}} \sum_{q \in Q} \frac{(q(x) - \hat{\mu})^2}{|\mathcal{X}||Q|}}, \qquad (8)$$

which can be computed easily with `np.mean` and `np.std`.

For fair comparisons, we use CLIP with `ViT-B/16` backbone. Each image is preprocessed with CLIP's default `preprocess` function (Radford et al., 2021).

**Architectures.** In this work, we only have two design choices for the predictor $f_\theta$ and querier $g_\eta$, which we will denote them as `shallow` and `deep`. `shallow` is a two-layer fully connected neural network, and `deep` is a four-layer fully connected neural network. Diagrams of their architecture are shown in Figure 11. The `shallow` architecture is used for medium-scale datasets CIFAR-10, CIFAR-100, CUB-200. The `deep` architecture is used for large-scale datasets ImageNet and Places365. The number of parameters for each experiment is listed in Table 3. The architecture design is chosen empirically based on the performance of the model. The size of the query set only affects the final output dimension of the querier $g_\eta$, and the number of classes affects the final output dimension of the classifier $f_\theta$. In all of our experiments, we do not share the weights between $f_\theta$ and $g_\eta$. We apply Softmax operator to class logits and query scores to obtain probability for each class and probability for each query. During training, a Straight-through softmax with temperature parameter $\tau$ is used for computing query probabilities. For every experiment, we linear decay $\tau$ from 1.0 to 0.2 for 20 epochs [2]. We find V-IP training is rather insensitive to how we anneal $\tau$.

---

[2] $\tau = 1.0$ is equivalent to regular Softmax operator, whereas $\tau \to 0$ corresponds to $\operatorname{argmax}(\cdot)$ operator.

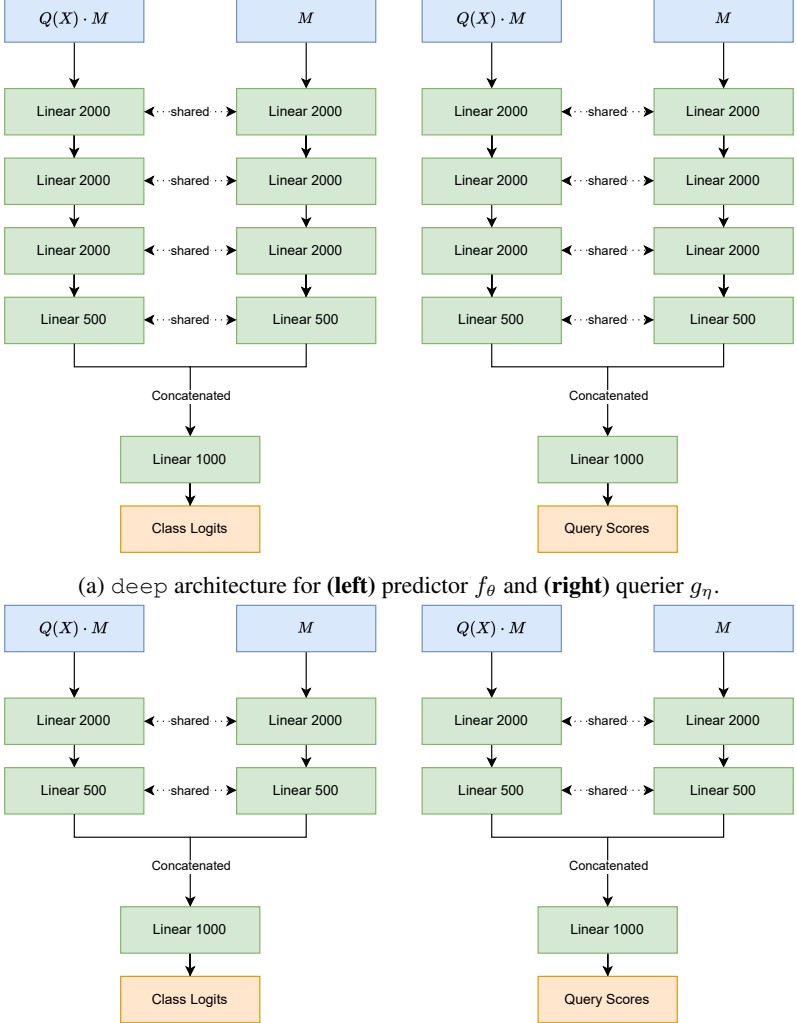

(a) `deep` architecture for **(left)** predictor $f_\theta$ and **(right)** querier $g_\eta$.

(b) `shallow` architecture for **(left)** predictor $f_\theta$ and **(right)** querier $g_\eta$.

Figure 11: Architecture designs for `shallow` and `deep`. "shared" implies the weights are shared between the two linear layers. "Concatenated" implies the output from previous layers are concatenated ($a \in \mathbb{R}^{1 \times n}, b \in \mathbb{R}^{1 \times}, \text{concat}(a, b) = [a|b] \in \mathbb{R}^{1 \times (n+m)}$). Every arrow $\rightarrow$ before the concatenation and after the input layer is LayerNorm of appropriate dimension, followed by ReLU.

| # of Parameters | CUB-200 | CIFAR-10 | CIFAR-100 | ImageNet | Places365 |
|---|---|---|---|---|---|
| CLIP+V-IP | 3,051,408 | 2,596,138 | 5,773,924 | 41,659,523 | 29,441,572 |

Table 3: The total number of parameters for each dataset.

**Optimization.** Every CLIP+V-IP experiment for CIFAR-10, CIFAR-100, CUB-200 follow the two-stage training procedure mentioned in the Chattopadhyay et al. (2023), where we optimize the V-IP objective for $f_\theta$ and $g_\eta$ with random histories, by first using the Random Sampling strategy for 4000 epochs with learning rate 0.0001, followed by the Subsequent Biased Sampling strategy for 1500 epochs with learning rate 0.00005. In both stages, we train using Adam Optimizer (Kingma & Ba, 2015) with no weight decay, along with Cosine Annealing learning rate scheduler and hyperparameter `T_max=100`.

Similarly, for large-scale datasets ImageNet and Places365, we also follow the two-stage training procedure. We use the same optimizer and learning rate scheduler with the same optimizer hyperparameter as mentioned above, but we train the Random Sampling stage for 400 epochs, and train the Subsequent Biased Sampling stage for 40 epochs. In both stages, we also train using Adam Optimizer (Kingma & Ba, 2015) with no weight decay, but with Cosine Annealing learning rate scheduler and hyperparameter `T_max=100`.

### E.2   TRAINING CLIP-BIN$_{\text{STD}}$+V-IP

The architecture and optimization exactly the same as described in §E.1 with the distinction that the query answers are now binarized by thresholding the standardized dot products from equation 8 by 0. A positive value indicates a "Yes" answer while a negative value indicates a "No" answer.

### E.3   TRAINING CLIP-BIN$_{\text{NORM}}$+V-IP

The architecture and optimization exactly the same as described in §E.1 with the distinction that the query answers are now binarized by thresholding $P_{\text{CLIP}}(C=1 \mid X=x)$ (equation 4) by 0.5. A higher value indicates a "Yes" answer while a lower value indicates a "No" answer.

### E.4   TRAINING CONCEPT-QA+V-IP

**Query Answers** $Q(X)$**.** Query answers are computed using a trained Concept-QA. To obtain binary query answers, we threshold the pre-sigmoid output logit of the Concept-QA, where the query answer is 1 (corresponding to a "Yes") if logit is above the threshold and 0 (corresponding to a "No") if logit is below. The threshold is -0.4 for CUB-200 and Places365 and 0 for others.

**Architecture.** The network architectures for V-IP with query answers from Concept-QA are shown in Figure 12. Similar to the querier architecture in Chattopadhyay et al. (2023), the querier is followed by a Straight-through Softmax, hence the output is a $|Q|$-dimensional one-hot vector.

Table 4: Number of training epochs during Random Sampling stage (RS) and Subsequent Biased Sampling stage (SBS). The maximum lengths of history allowed during SBS (denoted as Max. $|S|$) are also listed. $|Q|$ means for that dataset we used the size of the query set as the maximum allowed length of history for SBS.

| Dataset | RS | SBS | Max. $|S|$ |
|---|---|---|---|
| CUB-200 | 200 | 200 | $|Q|$ |
| CIFAR-10 | 200 | 100 | $|Q|$ |
| CIFAR-100 | 200 | 200 | $|Q|$ |
| Places365 | 1500 | 80 | 500 |
| ImageNet | 600 | 100 | 500 |

**Optimization.** For every dataset, training Concept-QA+V-IP follows the two-stage procedure as mentioned in the Chattopadhyay et al. (2023), where we optimize the V-IP objective for $f_\theta$ and $g_\eta$ with random histories. In the Random Sampling Stage, the classifier $f_\theta$ and the querier $g_\eta$ are

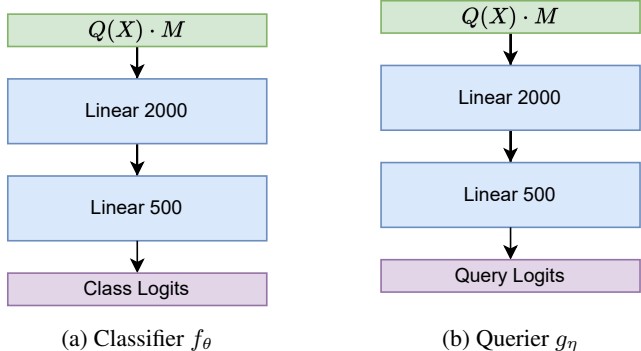

(a) Classifier $f_\theta$          (b) Querier $g_\eta$

Figure 12: Architecture for Concept-QA+V-IP. Between each linear layer is ReLU nonlinearity followed by LayerNorm.

optimized joining using Adam optimizer with learning rate 0.0001 and `amdgrad=True`, while learning rate is scheduled using Consine Annealing with `T_max=200`. In the Subsequent Biased Sampling stage, the classifier $f_\theta$ is optimized using SGD with learning rate 0.0001, Nesterov's momentum 0.9 and weight decay $5\cdot10^{-4}$, while learning rate is multi-step scheduled to decay at epochs with multiples of 30 by a factor of 0.2. On the other hand, querier $g_\eta$ is optimized with Adam with learning rate 0.0001 and no weight decay, while learning rate is scheduled with Cosine Annealing with `T_max=200`. Moreover, during Subsequent Biased Sampling, we limit the maximum length of the history for computational purposes. The number of epochs for each dataset during the two training phases, as well as the number of maximum length of the history allowed during training is listed in Table 4.

### E.5 TRAINING BLIP2-FLANT5+V-IP

The architecture and optimization exactly the same as described in §E.4 with the distinction that the query answers are now supplied by the pre-trained Vision-Language Model BLIP2 ViT-g FlanT5$_{XL}$.

### E.6 OBTAINING $P_{GPT}$ FROM LLAMA2

To obtain pseudo-labels from a language model Llama2, we provide the instruction prompt and a user prompt.

**Llama2.** We requested and downloaded the open-source weights for Llama model from Meta. We use the chat `llama-2-13b-chat` model in a zero-shot fashion (without any finetuning). The instruction prompt is as follows:

```
You are a helpful, respectful and honest assistant. Always
answer as helpfully as possible, while being safe.  Your answers
should not include any harmful, unethical, racist, sexist, toxic,
dangerous, or illegal content. Please ensure that your responses
are socially unbiased and positive in nature.
```

whereas the user prompt is the same as that for GPT from Appendix D. For hyperparameters, we set temperature to 0.6 and top-p probability for nucleus sampling to 0.9.

## F EVALUATING THE FAITHFULNESS OF OUR PSUEDO-LABELS

We evaluate the quality of our pseudo-labels by computing their accuracy and $F_1$-score of over the 5 image classification datasets considered in this paper. We also compare with baseline methods which are state-of-the-art VLMs that present an alternative approach to directly get pseudo-labels for training the Concept-QA model. Specifically, we compare with InstructBLIP (FlanT5$_{XL}$) (Dai et al., 2023) and the LLaVa-1.5$_{7B}$ models (Liu et al., 2024). For InstructBLIP (FlanT5$_{XL}$) we use the

same prompt as in footnote 1 of the main paper. For LLaVa-1.5$_{7B}$ we use the prompt "Is `concept` present in the image? Answer only Yes or No.", where `concept` is replaced by the name of the concept we are interested in asking the VLM about. Furthermore, we compare the quality of pseudo-labels generated when using a different LLM (LLaMa2 (Touvron et al., 2023)) instead of GPT in equation 5. For more details refer Appendix E.6.

We evaluate all four models on the same annotated data of image-concept pairs as used in Table 1. The results of this experiments are reported in Table 5 where "GPT+CLIP" refers to the pseudo-labels we used in training our Concept-QA method in this work (specifically equation 5).

Table 5: Results for evaluating the faithfulness of different pseudo-label generating mechanisms to the true answers. Acc. and F1 refer to accuracy and $F_1$-score metrics respectively.

| Pseudo-labelling method | ImageNet | | Places365 | | CUB-200 | | CIFAR-10 | | CIFAR-100 | |
|---|---|---|---|---|---|---|---|---|---|---|
| | Acc. | $F_1$ | Acc. | $F_1$ | Acc. | $F_1$ | Acc. | $F_1$ | Acc. | $F_1$ |
| GPT+CLIP | **0.86** | 0.55 | 0.82 | 0.42 | **0.79** | **0.53** | 0.81 | **0.62** | 0.79 | 0.37 |
| LLaMa2+CLIP | 0.81 | 0.46 | 0.75 | 0.47 | 0.62 | 0.45 | 0.78 | 0.61 | **0.84** | 0.33 |
| InstructBLIP (FlanT5$_{XL}$) | 0.85 | **0.58** | **0.86** | **0.66** | 0.68 | 0.46 | **0.82** | **0.62** | **0.84** | **0.40** |
| LLaVa-1.5$_{7B}$ | 0.63 | 0.41 | 0.66 | 0.47 | 0.49 | 0.46 | 0.69 | 0.52 | 0.66 | 0.25 |

Similarly, "LLaMa2+CLIP" refers to pseudo-labels generated using equation 5 when LLaMa2 is used as the LLM in place of GPT. From the results we make two observations. Firstly, we note that using GPT as the LLM provides far superior pseudo-labels than LLaMa2 on all the 5 datasets employed. Secondly, on all datasets our pseudo-labels (GPT+CLIP) are more faithful at representing the true ground truth answers than the LLaVa model while being only slightly behind when compared to InstructBLIP[3]

However, as discussed in §4 in the main paper, using these VLMs for training Concept-QA model is infeasible since they are computationally very intensive. For context, Imagenet has about 1.2 million images in its training set and the corresponding query set (obtained from GPT) has about 4.5K queries. In order to generate pseudo-labels for training our Concept-QA model we need to answer all queries about all images in the training data. This would require about 37K GPU hours using the InstructBLIP (FlanT5$_{XL}$) model (evaluated on NVIDIA RTX A5000 GPU, by averaging the time taken for computing a batch of 100 image-query pairs and then multiplying by the size of the training set and the query set) which is simply infeasible. In comparison, our GPT + CLIP methodology to generate psuedolabels (see equation 5) takes about 600 GPU hours. The bulk of this computation time comes from using GPT, which takes about 1-2 minutes to answer batches of 200 queries per class. Compared to the VLM, this is a massive speed-up! In Table 6 we report the time taken to generate pseudo-labels using our method vs. the time taken to annotate all image-query pairs using the InstructBLIP (FlanT5$_{XL}$) model for all 5 datasets employed. We do not compare with LLaVa, since our pseudo-labels outperform LLaVa on both metrics on all 5 datasets. However, since both LLaVa-1.5$_{7B}$ and InstructBLIP (FlanT5$_{XL}$) have the number of parameters of the same order (7B vs. 4B respectively), their evaluation times are of similar magnitudes.

Table 6: Time taken to annotated the entire training dataset with query answers using different pseudo-label generating mechanisms. Numbers in GPU hours.

| Dataset | GPT+CLIP (Ours) | InstructBLIP (FlanT5$_{XL}$) |
|---|---|---|
| CIFAR-10 | **0.2** | 41 |
| CIFAR-100 | **12** | 266 |
| CUB-200 | 20 | **6** |
| Places365 | **83** | 25,650 |
| ImageNet | **600** | 37,344 |

---

[3]with the exception of CUB-200 and Places365. On CUB-200, our pseudo-labels perform much better due to the domain-specific nature of this dataset which is too specialized for these general-purpose VLMs to have good zero-shot performance. On Places365, InstructBLIP achieves a much higher $F_1$-score than our pseudo-labels. We hypothesize that this is due to CLIP's image encoder not being able to capture essential image semantics for this dataset as discussed in §4 (discussion of results for Table 2.)

## G   ADDITIONAL EXAMPLES OF QUERY-ANSWER CHAINS

We illustrate more test examples of query-answer chains using ConceptQA+V-IP and CLIP+V-IP on CUB-200 13, CIFAR-10 17, CIFAR-100 16, ImageNet 14, Places365 15.For datasets with more than 10 classes, we show only the classes with the top 10 probability class where IP terminates. For examples using CLIP+V-IP, green/red query (y-axis of the plots) represents whether the query-answer is above/below 0. For examples using Concept-QA+V-IP, the green/red query represents whether query-answer is a yes/no. See caption of Figure 3 for a description of the heatmaps.

## H   LIMITATIONS & FUTURE DIRECTIONS

Following are the limitations of our work.

- Prior work on V-IP required the user to annotate datasets with query-answers which could be used to supervise the training of classifiers to provide answers at inference time. In this work, to overcome this data annotation bottleneck we propose to use GPT + CLIP to provide psuedo-labels to train our answering mechanism (the Concept-QA model). This inevitably leads to a loss in accuracy since we rely on the zero-shot performance of these large language models and vision language models. Future work would aim to address this limitation by exploring a hybrid approach where we manually annotate a small portion of the data and use it in tandem with the pseudo-labels to train the Concept-QA model.

- The current approach is limited to the problem of image classification due to the assumptions involved in deriving our pseudo-labels, that is, the class object is the focal point of the image (mentioned in Section 3.2.3). Future work would be aimed at exploring other ways of generating pseudo-labels to extend the application of V-IP to more general vision tasks such as image captioning, scene segmentation and visual reasoning to name a few.

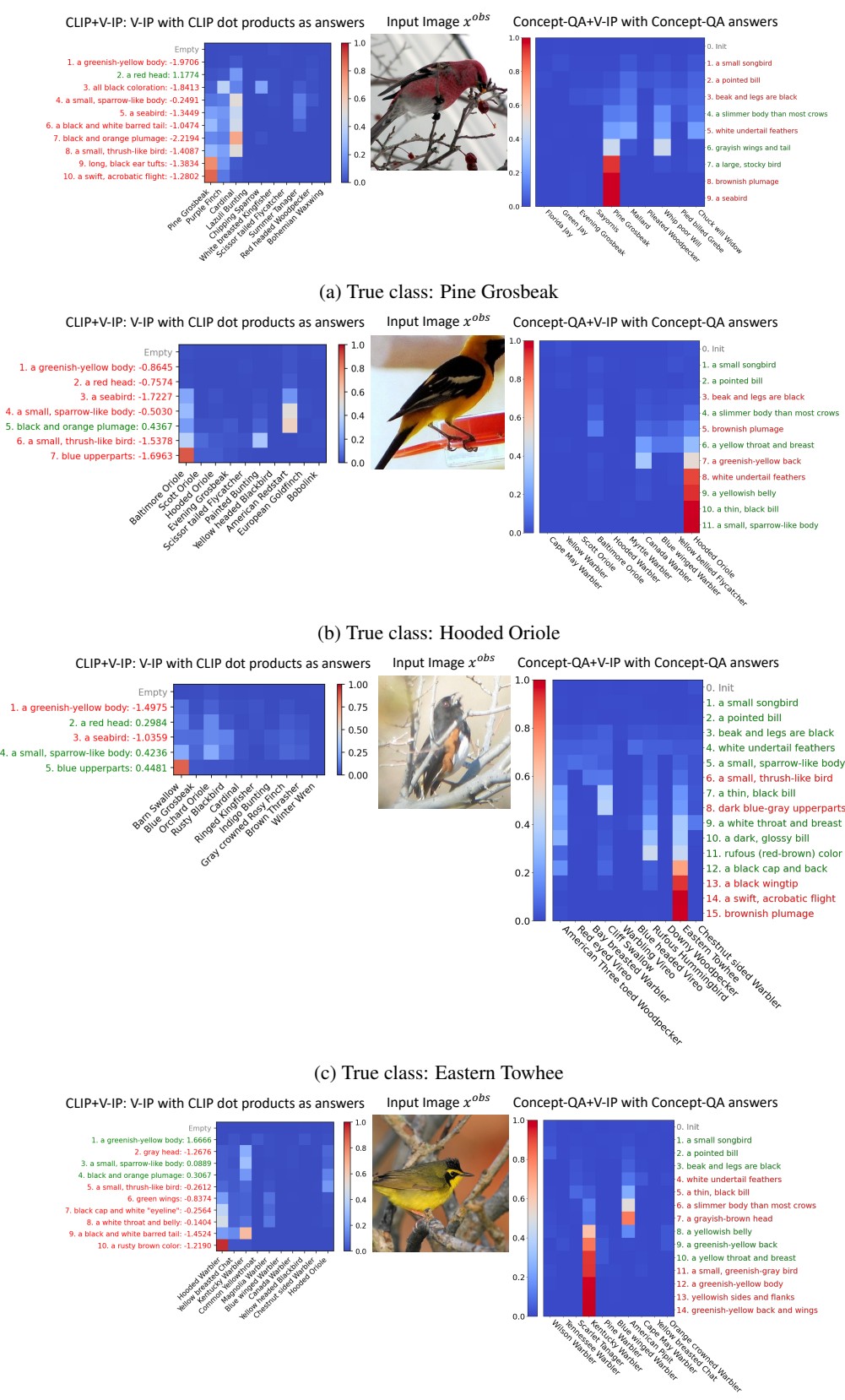

Figure 13: Additional Examples from CUB-200.

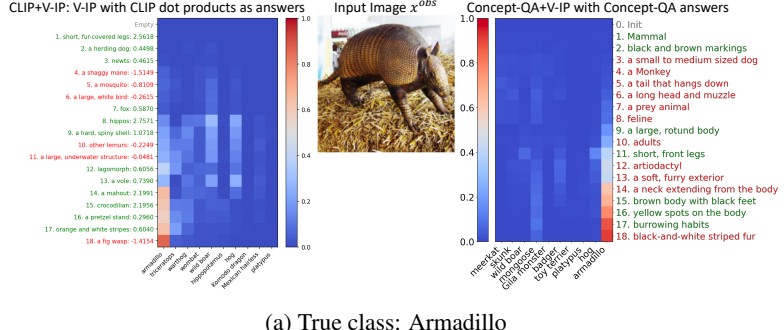

(a) True class: Armadillo

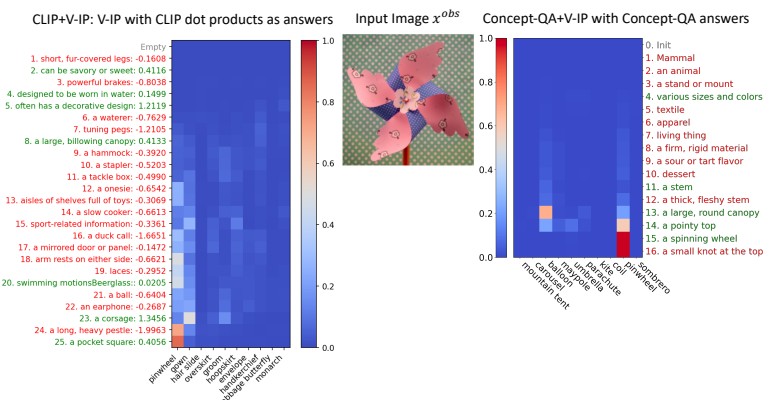

(b) True class: Pinwheel

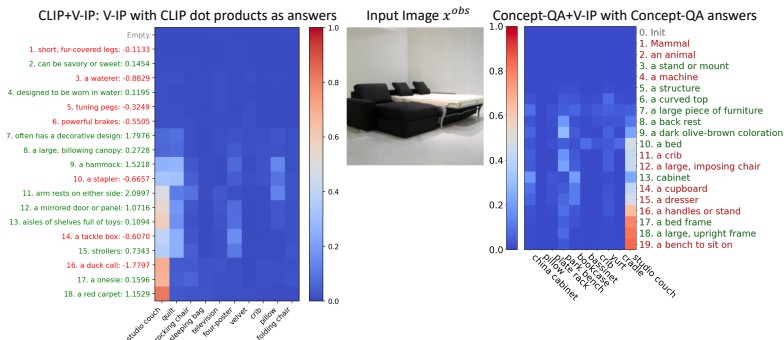

(c) True class: Studio Couch

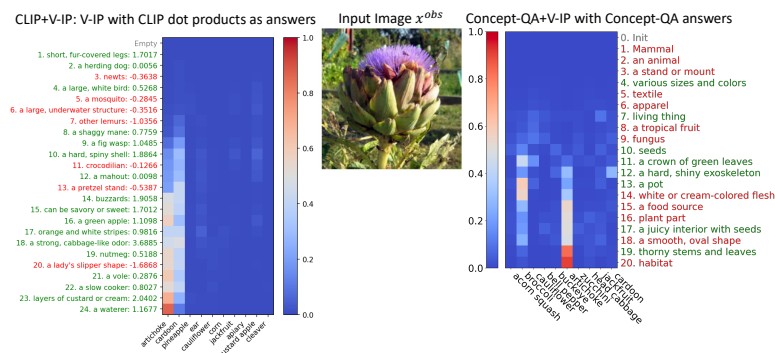

(d) True class: Artichoke

Figure 14: Additional Examples from ImageNet.

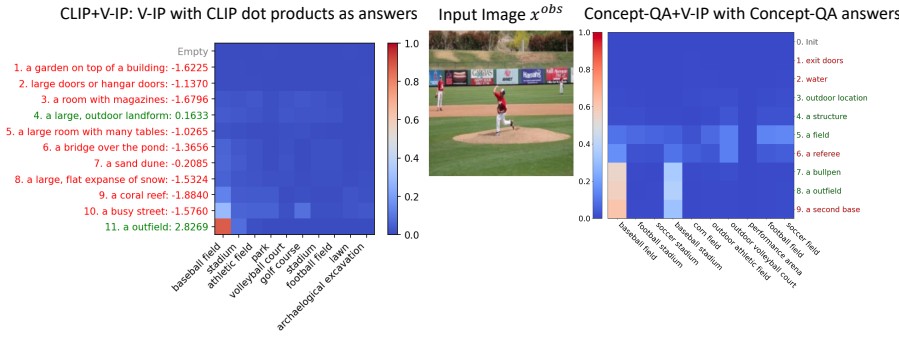

(a) True class: Baseball Field

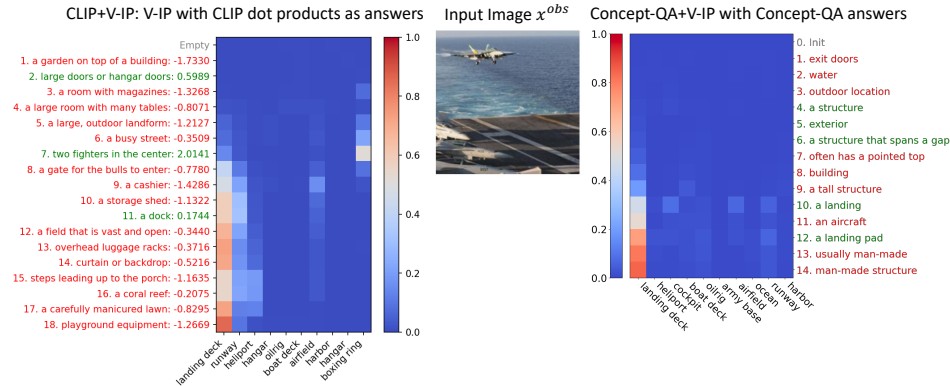

(b) True class: Landing Deck

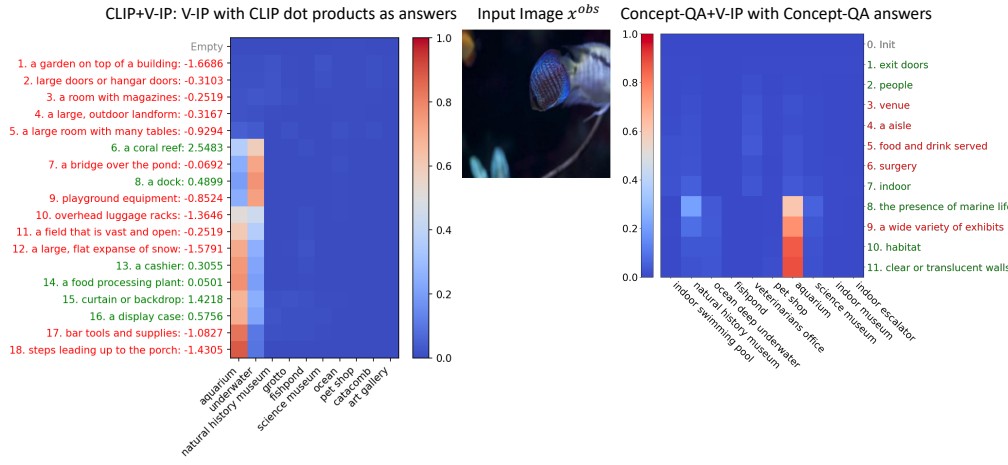

(c) True class: Aquarium

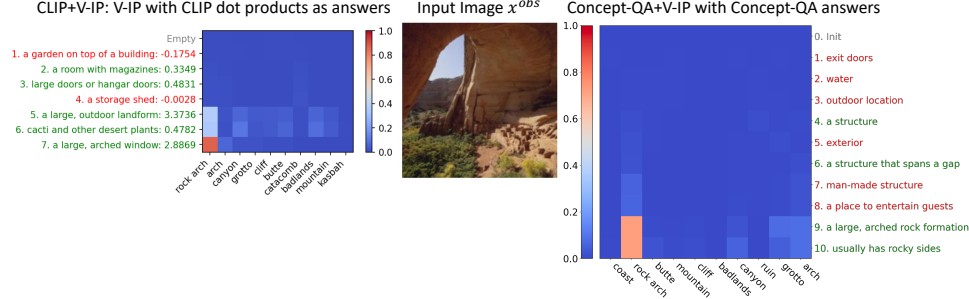

(d) True class: Roch Arch

Figure 15: Additional Examples from Places365.

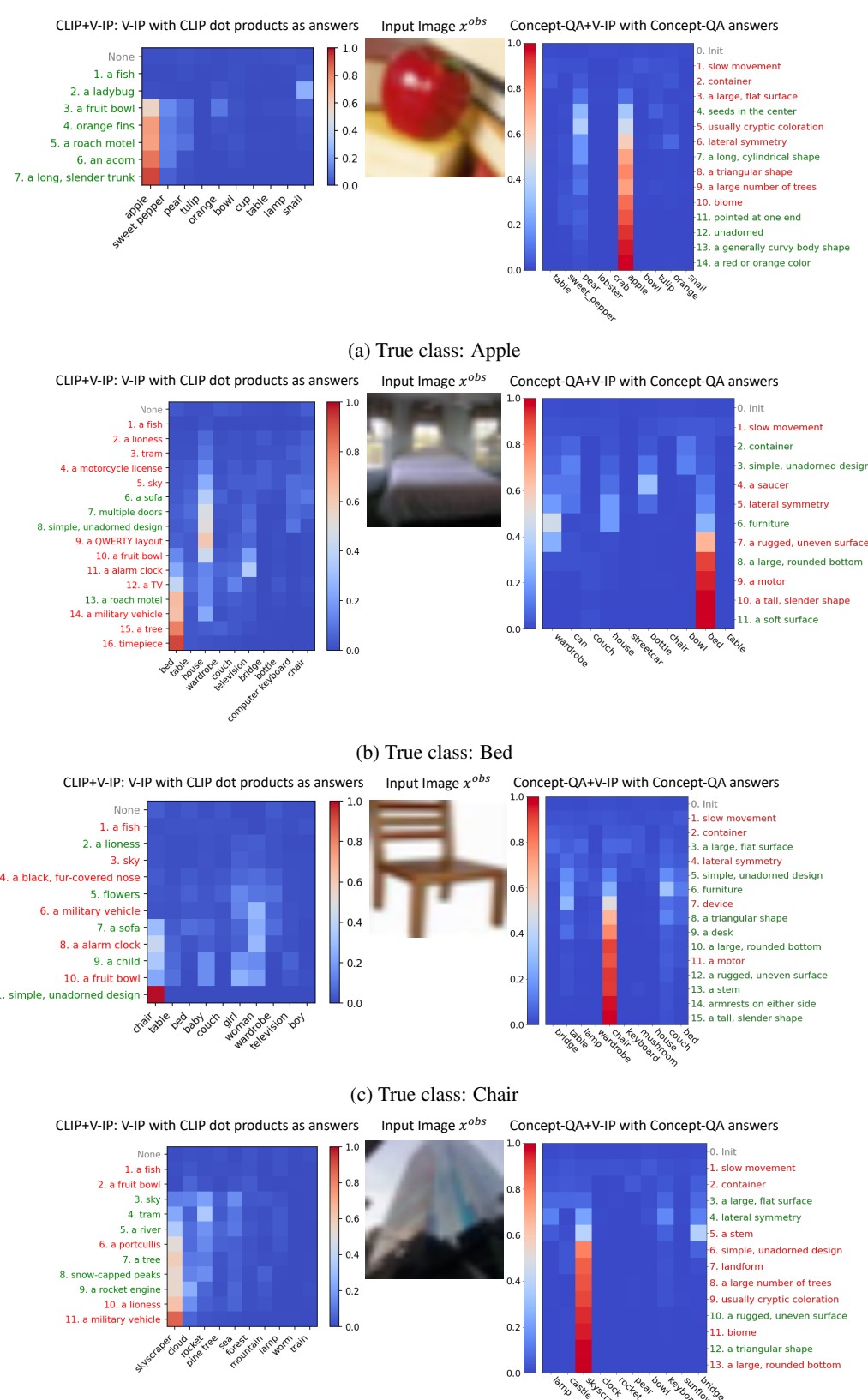

Figure 16: Additional Examples from CIFAR-100.

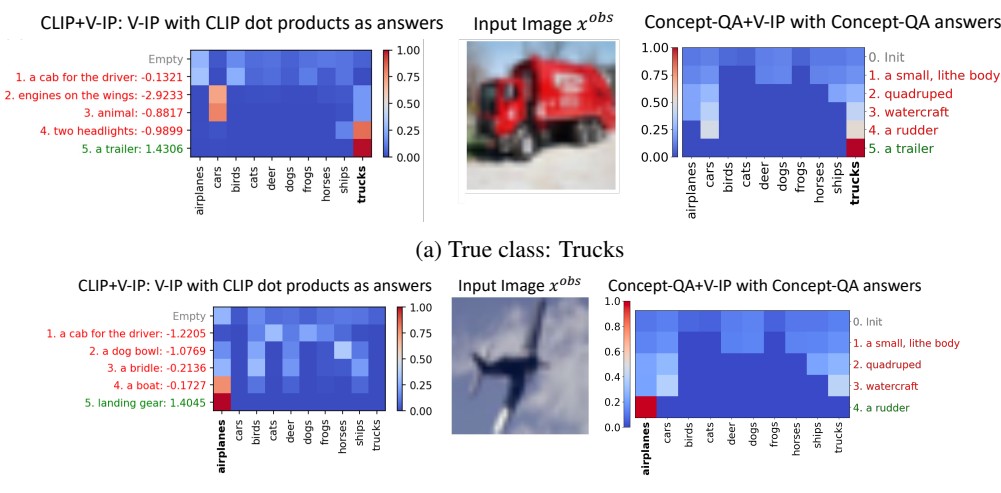

(a) True class: Trucks

(b) True class: Airplanes

Figure 17: Additional Examples from CIFAR-10.

