# OpenReview forum: "Bootstrapping Variational Information Pursuit with Large Language and Vision Models for Interpretable Image Classification"
_ICLR.cc/2024/Conference — ICLR 2024 poster_

### Official Review · Reviewer_3dNV · 2023-11-02

**Soundness:** 2 fair
**Presentation:** 3 good
**Contribution:** 3 good
**Rating:** 6
**Confidence:** 3

**Summary:**

This paper focuses on the problem of variational information pursuit (V-IP). The focus of V-IP is to generate the final answer by answering a series of interpretable queries/questions, facilitating the interpretability of model prediction. In this paper, the authors focus on a challenging problem of how to train a model to answer the interpretable queries as those queries are typically auto-generated and have no associated ground-truth labels. The authors thus proposed to generate pseudo-labels with the aid of GPT and CLIP.

The authors empirically show that the proposed Concept-QA model leads to shorter and more interpretable query-chains than the CLIP baseline.

**Strengths:**

- The paper is easy to follow. The authors explain their proposed method and give background knowledge about V-IP in an easy-to-follow way. More technical details are also explained and provided in the appendix.

- The designed experiments indeed show that the proposed Concept-QA helps the whole V-IP improve interpretability and results in a shorter query chain compared to the CLIP baseline.

- Fig 4 illustrates how using the proposed Concept-QA improves the interpretability of query chain.

**Weaknesses:**

- The major (if not the only) competitor of the proposed Concept-QA is CLIP baseline. It would be good to include other baseline methods for comparison.

- What the quality of the pseudo-label when evaluated on the Table 1 experiment?

- Since there exist many vision language models (commercial or open source), there might be other easier and more effective ways to generate pseudo-labels for training the Concept-QA model. For example, prompt a LLaVA or InstructBLIP model. The authors should compare to other pseudo-label generation methods as this is part of the major contribution of this paper.

- What's the correlation between the performance of the QA model and the final V-IP system? In the paper, the authors $\textbf{imply}$ that a better QA model leads to better overall V-IP performance, without supporting experiments and evidence. This weakens the connection between the results in Table 1 and the overall V-IP problem setting. For example, the author may show that using BLIP2 on Places 365 or CIFAR-100 brings better V-IP performance (improved interpretability of query chain, shorter query chain, etc). The reviewer is aware that the computation of BLIP2 is much more expensive than that of Concept-QA. The point here is to build the connection between QA model performance and V-IP performance.

- It would be good to come up with a way to quantify the interpretability and the length of query chain so that the reader can be more confident that the proposed Concept-QA indeed improves over CLIP baseline in these two aspects. For example, a few qualitative results in Fig 4 and the appendix might not be representative enough and might be cherry-picked.

**Questions:**

Please see the questions in the weakness section.

---

> ### Author Response · Authors · 2023-11-29
> **Response to Reviewer 3dNV (1/3)**
>
> Thank you for your time and feedback. We would respond to each of your comments (highlighted in bold and paraphrased) below.
>
> **The major (if not the only) competitor of the proposed Concept-QA is CLIP baseline. It would be good to include other baseline methods for comparison.**
>
> In response, we have added more baselines. In Table 1 (which compares the accuracy of Concept-QA answers with baseline) we have added a comparison with binarized CLIP answers by thresholding at 0.5 after the min-max normalization of the dot products along with the original comparison with 2 state-of-the-art vision-language models and CLIP answers binarized by thresholding at 0 after standardization of the dot products. Moreover, in Figure 5, we have added accuracy vs. average number of queries needed curves for the following new baselines: V-IP trained with BLIP2 (a large vision-language model), V-IP trained with CLIP binarized by thresholding at 0 after standardization of the dot products, and V-IP trained with CLIP binarized by thresholding at 0.5 after min-max normalization of the dot products.
>
> **What the quality of the pseudo-label when evaluated on the Table 1 experiment?**
>
> We have done this experiment and reported the results in Table 5 in the Appendix. Here is a comparison with Concept-QA which was trained using these psuedo-labels.
>
> | Model         | Imagenet      | Places365     | CUB-200       | CIFAR-10      | CIFAR-100    |
> |---------------|---------------|---------------|---------------|---------------|--------------|
> |               | Acc \| F1     | Acc \| F1     | Acc \| F1     | Acc \| F1     | Acc \| F1    |
> | Concept-QA    | 0.87 \| 0.56  | 0.83 \| 0.45  | 0.80 \| 0.54  | 0.80 \| 0.62  | 0.80 \| 0.38 |
> | Pseudo-labels | 0.86 \| 0.55  | 0.82 \| 0.42  | 0.79 \| 0.53  | 0.81 \| 0.62  | 0.79 \| 0.37 |
>
>
> Our results show a close agreement between the correctness of our pseudo-labels and Concept-QA (which indicates that Concept-QA is able to learn effectively from the pseudo-labels).
>
> **Since there exist many vision language models (commercial or open source), there might be other easier and more effective ways to generate pseudo-labels for training the Concept-QA model. For example, prompt a LLaVA or InstructBLIP model. The authors should compare to other pseudo-label generation methods as this is part of the major contribution of this paper.**
>
> Thank you for this suggestion. We have made this comparison and reported results in Appendix F of the updated paper. In particular, our pseudo-labels outperform Llava on all datasets and are competitive with InstructBLIP. More specifically, for InstructBLIP, our results are competitive for Cifar-10, Cifar-100 and Imagenet datasets. On CUB-200 our psuedolabels are better and we believe this is due to the domain-specific nature of this dataset. On Places365 on the other hand, InstructBLIP answers perform better (with respect to the ground truth) and we attribute this to the CLIP image encoder's inability to capture salient features for this dataset (as evidenced by low accuracy of training a linear probe on top of CLIP's image representation for Places365 (Table 2)).
>
> However, we want to re-emphasize that the main bottleneck for using these large vision-language models is their computation cost. Passing a single image through InstructBLIP to get an answer takes about 1-2 seconds on an NVIDIA RTX A5000 GPU, and thus it is infeasible to use InstructBLIP to annotate data for training Concept-QA. This point has been discussed in detail in Appendix F in the updated paper. Table 6 provides a comparison of the time taken to annotate query answers (for training Concept-QA) using our method of generative pseudo-labels vs. the time taken to annotate using the InstructBLIP model. As an example, to obtain annotations for the entire Imagenet training data, it would take about 600 GPU hours using our method vs. 37,344 GPU hours using InstructBLIP (all evaluated on NVIDIA RTX A5000 GPU).

---

> ### Author Response · Authors · 2023-11-30
> **Response to Reviewer 3dNV (2/3)**
>
> **What's the correlation between the performance of the QA model and the final V-IP system? In the paper, the authors imply that a better QA model leads to better overall V-IP performance, without supporting experiments and evidence. This weakens the connection between the results in Table 1 and the overall V-IP problem setting.**
>
> The evaluation of the QA model and the V-IP system are meant to evaluate two complementary objectives for the overall framework and there need not be a correlation between them. We expand on this below.
>
> 1. The individual queries in the query set must have a clear interpretation to the user (since the V-IP explanations are a conjunction of queries selected from this query set along with their obtained answers). The first set of experiments (see Table 1) is aimed at evaluating the interpretability of the GPT-proposed query set. This is important since Concept-QA is trained using pseudo-labels, thus if for the question ”Does it have stripes?”, Concept-QA answers ”yes” when the image has no stripes but has a checkerboard pattern (a completely different concept) then this query isn’t well-defined (from an interpretability point of view).
>
> 2. Assuming every query in the defined query set is interpretable, shorter explanations are easier to parse and understand than longer ones assuming both explanations are “sufficient” (which necessarily means that the longer explanation contains redundant information). This “sufficiency” of explanations is captured by the test accuracy. Stated precisely, if two strategies have the same classification accuracy, then we prefer the strategy that has a smaller explanation length on average (as measured by the number of queries asked by V-IP). This is based on Occam’s Razor, which is one of the most widely accepted heuristics used in science to choose between competing explanations for the same phenomena. The next set of experiments (reported in Figure 5) is aimed at evaluating this in terms of the trade-off between test accuracy and average explanation length.
>
> We have modified the opening of the experiments section in the updated paper (Section 4) to better motivate our design of experiments (as discussed above).
>
> Finally, we would like to point out to the reviewer that there need not be a correlation between the two above points and we have **not** implied this in the paper. In a sense, one could have a very interpretable query set, with the QA system being highly accurate, but if the query set is not sufficient or appropriate for the classification task then V-IP would not be able to get good a classification accuracy. For example, say you only have queries about the colour of birds, nothing about other characteristics like beak shape, size, and body shape. Even a perfectly accurate QA system for the colour of birds would not provide enough information for V-IP to classify the CUB-200 dataset with high accuracy. Similarly, interpretability is not needed for high classification accuracy. Case in point: deep networks are state-of-the-art in most vision tasks but are widely accepted to be uninterpretable black boxes.
>
> **It would be good to come up with a way to quantify the interpretability and the length of query chain so that the reader can be more confident that the proposed Concept-QA indeed improves over CLIP baseline in these two aspects.**
>
> Please refer to our answer for the previous comment. The philosophy of the V-IP framework (discussed in the original paper which introduced this framework, Chattopadhyay et al., TPAMI 2022) is as follows: First an interpretable query set is defined where every query has a well-defined interpretation to the user, and then V-IP is used to compose the queries into short chains to explain the prediction. By design, any chain (which is simply a conjunction of interpretable queries) is interpretable. Second, shorter chains are just easier to parse (takes less effort) and interpret than longer ones, which is in accordance with the popular Occam's razor principle. Thus, Table 1 shows that the proposed Concept-QA produces a more ``interpretable" query set than CLIP baselines since they produce more accurate query answers (with respect to the ground truth answers) than baseline models and Figure 5 shows that V-IP with Concept-QA as the answering mechanism achieves shorter chains with higher accuracy than V-IP with baseline answering mechanisms.

---

> ### Author Response · Authors · 2023-11-30
> **Response to Reviewer 3dNV (3/3)**
>
> **The author may show that using BLIP2 on Places 365 or CIFAR-100 brings better V-IP performance (improved interpretability of query chain, shorter query chain, etc). The reviewer is aware that the computation of BLIP2 is much more expensive than that of Concept-QA. The point here is to build the connection between QA model performance and V-IP performance.**
>
> Thank you for this suggestion, indeed using BLIP2 in conjunction with V-IP is too expensive to train on some of the larger datasets (Imagenet and Places365) as the reviewer commented. Instead, we have reported the query chain length vs. accuracy trade-off for BLIP2 on the three smaller datasets, namely CIFAR-10, CIFAR-100 and CUB-200. These results are reported in Figure 5 of the updated paper. Our result indicates that Concept-QA+V-IP outperforms BLIP2-FlanT5+V-IP on CIFAR-10 and CUB-200 (the latter by a huge margin). On CIFAR-100, our model has a better trade-off for short explanation lengths but eventually the BLIP2-FlanT5+V-IP gets better results.
>
> However, as stated in previous responses there need not be a correlation between QA model performance and V-IP performance, but the fact that a better QA model results in better V-IP performance in our experiments is certainly serendipitous!

---

### Official Review · Reviewer_ZeKa · 2023-11-04

**Soundness:** 3 good
**Presentation:** 3 good
**Contribution:** 2 fair
**Rating:** 5
**Confidence:** 4

**Summary:**

Background: Variational Information Pursuit is an approach to generate interpretable explanations that leverages densely annotated data.

Key idea: Overcome the bottleneck of dense-human-labeling by automatic labeling using an LLM.

Approach: Start from an existing list of “semantic concepts” (queries) generated by GPT. Get pseudo-labels generated from CLIP + GPT. QA model provides the answers. The QA model is trained on the GPT queries and  Together the query-answer pairs serve as explanations.

Contributions:

- Lightweight “Concept-QA” model
- Generated explanations (query-chains) are shorter, more accurate than with baseline QA model (CLIP-similarity score).

**Strengths:**

1. Performance: Concept-QA appears to perform well when evaluated along multiple axes. The results are highly correlated with concepts that are actually present in the images; i.e., there seems to be very little confabulation (often called “hallucination”). In addition, the explanations (query-answer chains) are shorter and more human-interpretable.

2. The key contribution of the paper appears to be the formulation of the Concept-QA model based on query information and answers from GPT + CLIP. The paper demonstrates that naively using CLIP-score between (query-concept, image) does not work well out-of-the-box, and proposes learning a new light-weight network based on pseudo-labels.

3. Impact: The labeling requirement was a huge bottleneck. With this approach, that requirement doesn’t exist anymore. It paves the way for more widespread application of VIP in scenarios where interpretable-by-design approaches are critical. However, we should credit the core idea and (part of the implementation) to the earlier work on label-free CBMs.

**Weaknesses:**

1. The framework of Variational Information Pursuit is quite similar to the Concept Bottleneck Models. The main difference is the sequential selection of informative queries (concepts) in VIP. The key contribution of the paper is the approach to overcome the limitation of annotating  query sets and labels. However, this was already proposed and implemented by an earlier paper (Oikarinen et al.).

2. Sec. 3.2.2 presents the observation that choosing a set of queries from the dataset a-priori (agnostic to the image), does not result in either an optimal or interpretable query set. This is understandable, since two main information sources are ignored — the image content, and taking advantage of the answers to the queries. Considering the actual experimental setting is significantly different from the VIP setting, the title “Answering Queries with CLIP Adversely affects VIPs explanations” and the conclusions seem a bit misleading. While floating-point-dot-product-scores from CLIP might also be a problem (as discussed in Sec. 4.1), the more obvious problem in this setting appears to be the query selection strategy. Please let me know if I have misunderstood something.

3. The proposed ConceptQA architecture is intuitive and quite straightforward. While this is not a downside by itself, it is probably something that one implement as a baseline. It is a bit hard to identify the interestingness or novelty in the approach. The closest baseline is simply a comparison against a CLIP(image, concept) similarity score.

References:
- Oikarinen et al.: Label-Free Concept Bottleneck Models; ICLR 2023.

**Questions:**

1. The overall approach is inspired by work on label-free CBMs and the important modules (query generation, answer-generation and concept-qa) are reasonable. However, it is a bit difficult for me to articulate precisely what are the novel scientific ideas or findings from this paper. It might be nice if the authors could reiterate this.

2. The main baseline for the proposed Concept QA model is binarized CLIP after normalization. There’s an assumption here that the value of “0” is a good threshold. Were there experiments performed to identify the optimal similarity threshold for accurate classification? For instance, the min-max normalization described in 3.2.3 might be a good starting point.

---

> ### Author Response · Authors · 2023-11-25
> **Response to Reviewer ZeKa (1/3)**
>
> Thank you for your time and feedback. We would respond to each of your comments (highlighted in bold and paraphrased) below.
>
> **Impact: The labeling requirement was a huge bottleneck. With this approach, that requirement doesn’t exist anymore. It paves the way for more widespread application of VIP in scenarios where interpretable-by-design approaches are critical. However, we should credit the core idea and (part of the implementation) to the earlier work on label-free CBMs.**
>
> We are glad the reviewer appreciates our contribution to ameliorating the labelling bottleneck for V-IP. We agree our idea was inspired by the work on label-free CBMs and a few others and we have cited them appropriately at several places in the main text of our original submission. For example, in the fourth paragraph of the introduction.
>
> **The framework of Variational Information Pursuit is quite similar to the Concept Bottleneck Models. The main difference is the sequential selection of informative queries (concepts) in V-IP.**
>
>  While both Variational Information Pursuit (V-IP) and Concept Bottleneck Models (CBMs) are types of explainable machine learning algorithms, the nature of explanations provided by both these methods is fundamentally different. This has been discussed in detail in the second paragraph of Section 2 in the main paper (related works). We summarize this briefly in the next paragraph.
>
> The sequential selection of queries in V-IP results in a progressive explanation of the final prediction akin to the popular parlour game 20 questions. The first ``most informative" query selected is independent of the input image. Depending on the answer to this query, the second query is selected and so on. The sequence of query-answer chain selected until prediction serves as an explanation for the prediction. Moreover, in each iteration, we observe the model's posterior change due to the query-answer pairs obtained so far, which adds a layer of transparency to the model's decision-making process. Such a progressive description of the model's decision-making process is not available in CBMs which typically use a linear network to predict class labels from concepts.
>
> **The key contribution of the paper is the approach to overcome the limitation of annotating query sets and labels. However, this was already proposed and implemented by an earlier paper (Oikarinen et al.).**
>
> We disagree with the reviewer. It is true that the key contribution of this work, as articulated by the reviewer, is to overcome the limitation of annotating query sets, thereby enabling the more widespread application of V-IP. However, in our opinion, the previous proposal of Oikarinen et al. for annotating query sets with CLIP dot products is not an acceptable solution since it makes V-IP's trajectories uninterpretable. To put it simply, continuous-valued answers are hard to interpret. For example, CLIP's dot product answer of $0.24$ to the question ``Is this a mammal?" does not have a clear interpretation. Is the image of a mammal or not? This in turn makes it harder to interpret the rationale behind the selection of subsequent queries which depends on the query-answers obtained so far. This is discussed in detail with examples in Section 3.2.2 of the main text.
>
> Our main contribution is to show that one can effectively train a Concept-QA model (by using CLIP and GPT to provide pseudolabels for training) to produce discrete binary answers which we empirically evaluate to produce more accurate predictions when used in conjunction with V-IP over the CLIP dot-products solution proposed by (Oikarinen et al.). Moreover, we show that the V-IP explanations generated using Concept-QA binary answers are interpretable and constrast them with the uninterpretable sequences obtained using CLIP (experiments section of the paper).

---

> ### Author Response · Authors · 2023-11-25
> **Response to Reviewer Zeka (2/3)**
>
> **Sec. 3.2.2 presents the observation that choosing a set of queries from the dataset a-priori (agnostic to the image), does not result in either an optimal or interpretable query set. This is understandable, since two main information sources are ignored — the image content, and taking advantage of the answers to the queries. Considering the actual experimental setting is significantly different from the VIP setting, the title “Answering Queries with CLIP Adversely affects VIPs explanations” and the conclusions seem a bit misleading. While floating-point-dot-product-scores from CLIP might also be a problem (as discussed in Sec. 4.1), the more obvious problem in this setting appears to be the query selection strategy. Please let me know if I have misunderstood something.**
>
> This is a misunderstanding. The set of queries chosen are not selected a-priori agnostic to the image. They are obtained by carrying out inference using V-IP trained on Imagenet. Specifically, we first train V-IP's querier and predictor networks on the Imagenet dataset using the query set used by Oikarinen et al (2023) and use CLIP's dot product to answer the queries. The example in Figure 2 (Figure 3 in the updated paper), shows inference by V-IP on a validation image from the Imagenet dataset. The queries selected are the most informative query in each step, as dictated by the trained V-IP's querier, while the heatmap shows how the model's posterior over the class labels change with each iteration. Thus, the actual experiment in Sec. 3.2.2 is same as the V-IP setting and is meant to illustrate how floating-point-dot-product-scores from CLIP are not interpretable. Consequently, the query-answer chain obtained using CLIP dot products as answers seem mysterious to the user as an explanation of the prediction. We have modified the third paragraph in Sec. 3.2.2 clarifying this. More examples contrasting the query-answer sequences produced by V-IP when trained using Concept-QA vs CLIP continuous-valued dot products is available in Figure 5 and Figures 13-17 in the Appendix.
>
>
> **The overall approach is inspired by work on label-free CBMs and the important modules (query generation, answer-generation and concept-qa) are reasonable. However, it is a bit difficult for me to articulate precisely what are the novel scientific ideas or findings from this paper. It might be nice if the authors could reiterate this.**
>
> The main scientific ideas and/or findings in this paper as as follows:
>
> 1. Our paper raises a key issue about the uninterpretability of V-IP's explanations when continuous-valued functions are used a answers, for example, dot products supplied by CLIP. To the best of our knowledge, this issue has not been discussed in the prior literature. We believe this is a critical point that prevents naively using prior work on label-free CBMs as the answering mechanism for V-IP.
>
> 2. Subsequently, we propose a solution, Concept-QA, to train a query-answering mechanism which provides discrete (binary) answers to queries. We then empirically show how Concept-QA resolves the interpretability issue of continuous-valued dot products.
>
> 3. We then successfully train V-IP with Concept-QA as the answering mechanism to extend the applicability of V-IP to other datasets where human-annotated query sets are not available. For example, we can now apply V-IP for the first time on the Imagenet dataset, which is arguably one of the most well-established benchmark datasets for image classification.
>
> **The main baseline for the proposed Concept QA model is binarized CLIP after normalization. Other baselines such as, the min-max normalization described in 3.2.3 might be a good starting point.**
>
> Thank you for this suggestion. We have added more baselines to compare with the proposed Concept-QA model. To summarize, we have now compared with (i) CLIP continuous-valued dot products after standardization (subtract and divide by the mean and std computed across all image-concept pairs in the training dataset), (ii) binarized CLIP by thresholding the (standardized) dot-products at '0', and (iii) binarized CLIP by thresholding the dot products after min-max normalization at `0.5'. We trained V-IP using each of these baselines as a potential answering mechanism and compared their corresponding test accuracy vs. the average number of queries needed curves across various image classification datasets (Figure 5). Our experiments show that V-IP with Concept-QA as an answering mechanism is superior to all versions of CLIP considered in the sense that they require a smaller average number of queries (shorter description length is preferred for interpretability) to achieve the same test accuracy (except for a small “low test accuracy, low avg. number of queries” regime on Cifar-10). Moreover, Table 1 shows that Concept-QA is more faithful to the true answer than these baselines. These updated results can be found in the experiments section of the updated paper.

---

> ### Author Response · Authors · 2023-11-25
> **Response to Reviewer ZeKa (3/3)**
>
> **There’s an assumption here that the value of “0” is a good threshold for binarizing the CLIP dot products after standardization. Were there experiments performed to identify the optimal similarity threshold for accurate classification?**
>
> No, we have not experimented with different binarizing thresholds to find the optimal value for classification for the following reasons.
>
> 1. We empirically find that V-IP trained with Concept-QA as an answering mechanism outperforms (in terms of the avg. number of queries needed to reach a certain level of test accuracy) V-IP trained with CLIP dot products (continous-valued) as an answering mechanism on all 5 datasets considered in this paper (see Figure 5). Since any sort of binarization of the continuous dot products would result in loss of information (this argument can be made rigorous using the data processing inequality), we expect V-IP's performance using CLIP dot products to be an upper bound on the performance one can obtain using any sort of binarization scheme (in particular any choice of threshold). This claim is substantiated by results in Figure 5, which clearly show that the curves produced by continuous-valued CLIP answers are superior to both binarization schemes considered in terms of the avg. number of queries needed to reach a particular level of test accuracy.
> 3. We believe the choice of using `0' is a natural and reasonable choice for binarization since this aligns with the commonly associated interpretation to CLIP dot products for grounding concepts to images where a positive value indicates presence of concept in the image and a negative value indicates its absence. In fact, this was the convention followed by Oikarinen et al. ICLR 2023, the reference provided by the reviewer.

---

### Official Review · Reviewer_8mpd · 2023-11-21

**Soundness:** 3 good
**Presentation:** 2 fair
**Contribution:** 2 fair
**Rating:** 6
**Confidence:** 4

**Summary:**

The focus of this paper is to revisit "Variational Information Pursuit (VIP) for image classification. VIP is an interpretable framework that makes predictions by sequentially selecting Question-Answer chains to reach the prediction. The paper states that VIP has limitations as it requires specification of a query set and densely annotated sequential QA data.  To relieve this limitation, the paper proposes that language models could be leveraged to generate queries for image classification, and a QA network is proposed (ConceptQA) to answer binary questions about the image.

Findings: Experiments are conducted on 5 image classification datasets and compared to 3 vision-language models (VLMs). The model outperforms the baselines on 2/5 datasets and is second best on the others.

**Strengths:**

1. This is an interesting take to use VIP / sequential QA for image classification and the experiments demonstrate the effectiveness of this method for interpretability/explanations.
2. In general the idea could be useful beyond image classification for other tasks with data-scarce settings, especially unlabeled datasets, where language models can help to create pseudo labels.
3. The paper put together (main + appendix) is well written and explains the preliminaries, proposed method, implementation, and experiments in good detail.

**Weaknesses:**

1. Comparisons to vision-only models are missing: it would be useful to know how much of a gap there is between supervised as well as self-supervised vision models trained on these datasets. This comparison would tell us how much performance is being sacrificed for interpretability. Comparisons in terms of #parameters are also important in this regard.
2. The method only uses OpenAI's GPT models (which are not open-source but only accessible via API calls -- i.e. need an internet connection and OpenAI account for inference) -- it would have been better to also implement the method with a local language model -- it is understandable that the performance could potentially be lower than GPT.
3. A human study could have helped to supplement the study on explanation length. See Q2 and Q3 below.
4. Useful details and useful visualizations are relegated to the appendix -- for instance Fig 9 could be moved to the main paper, some of the details on query set generation (App C) and training process (App D) could be briefly added in the main paper to improve the flow/readability of the paper.

**Questions:**

1. Given that the method works by using QA -- could it also be useful for improving VQA performance (eg. for VQA-v2, VizWiz, GQA, CLEVR etc datasets).
2. Do humans prefer shorter or longer explanations when it comes to trusting a model? While the tradeoff in figure 5 is a good finding, I'm not sure how it translates to helping humans understand decisions made by the model.
3. Similarly, when humans know that the prediction has failed -- do they prefer shorter or longer explanations to debug why it failed? This study could be added.

Comment:
1. I'm going to be "that" person and say that we should really avoid the marketing lingo of "Foundation models" in scientific papers -- who gets to decide which model is a foundation model and which isn't? Why is CLIP a foundation model but ResNet / RNN isn't?  The term "Large" language model (like "deep" learning) is also dicey, but at least we could define "large"/"deep" in terms of some threshold on number of trainable parameters/layers in the model. Is there really a definition for foundation models -- or is it just a marketing trick that everyone is accepting? I would recommend replacing "Foundation Models" in the title with "Language Models" -- it is more informative (because the paper relies on prompting GPT with natural language)

---

> ### Author Response · Authors · 2023-11-27
> **Response to Reviewer 8mpd (1/2)**
>
> Thank you for your time and feedback. We would respond to each of your comments (highlighted in bold and paraphrased) below.
>
> **Comparisons to vision-only models are missing: it would be useful to know how much of a gap there is between supervised as well as self-supervised vision models trained on these datasets. This comparison would tell us how much performance is being sacrificed for interpretability. Comparisons in terms of #parameters are also important in this regard.**
>
> We had already done a comparison and reported the results in our original submission. We believe the reviewer might have missed this. More specifically, this wass the last sentence of our main text - “Finally, since interpretability can be seen as a constraint on learning, we discuss in Appendix section A.1, the gap between V-IP’s performance using query sets to that of a black-box model.” In the updated paper this line appears as the last line of the second paragraph on page 9. The results appear in Table 2 of the Appendix.
>
> In particular, we compared the accuracy Concept-QA+V-IP obtained using the proposed GPT query sets vs. the accuracy of a linear classifier on top of CLIP’s ViT-B/16 image encoder (ViT-B/16 was the CLIP encoder used in our paper for providing pseudo-labels for training Concept-QA and also to produce the image encodings which go as input to the Concept-QA model). According to the reviewer’s recommendation, this can be considered as a vision-only baseline. Our results indicate that Concept-QA+V-IP is able to achieve a test accuracy close to this baseline on most of the datasets with an appreciable gap in case of CUB-200. We have also added another column to Table 2 (in the updated paper) indicating the accuracy ViT-B/16 (supervised training) achieves on these datasets as the supervised vision-only baseline requested by the reader.
>
> Lastly, we believe comparing the same vision transformer (ViT-B/16) which was used as a backbone for generating image embeddings to Concept-QA+V-IP ensures the comparison is fair and that all models considered have parameters of the same order (about 100-180M).
>
> **Avoid the marketing lingo of "Foundation models" in scientific papers.**
>
> We agree with the reviewer. We have changed the term "Foundation Models" in the title to "Large Language and Vision Models". This choice is made to incorporate both GPT (a language model) and CLIP (a vision language model).
>
> **Given that the method works by using QA – could it also be useful for improving VQA performance (eg. for VQA-v2, VizWiz, GQA, CLEVR etc datasets).**
>
> We hope our work motivates future investigations into using language models to construct pseudo-labels for more general vision tasks including VQA. However, in it’s current formulation we do not expect it to perform well since our pseudo-labels relies on the assumption that the class object is the focal point of the image (Section 3.2.3, third paragraph). This assumption generally holds for image classification problems but not for more general vision tasks.
>
> **Useful details and useful visualizations (for example figure 9) are relegated to the appendix that can be moved to the main paper to improve the flow/readability of the paper.**
>
> Thank you for this suggestion. We have moved Figure 9 to the main paper as suggested.

---

> ### Author Response · Authors · 2023-11-27
> **Response to Reviewer 8mpd (2/2)**
>
> **The method only uses OpenAI’s GPT models (which are not open-source but only accessible via API calls – i.e. need an internet connection and OpenAI account for inference) – it would have been better to also implement the method with a local language model – it is understandable that the performance could potentially be lower than GPT.**
>
> Thank you for this suggestion. Due to time constraints of the rebuttal we are unable to perform these experiments as of now. However, we would be happy to add this in the final version of the paper. Since GPT is the current state-of-the-art language model we expect the performance of other local LLMs to be comparatively lower (as expected by the reviewer).
>
> **Do humans prefer shorter or longer explanations when it comes to trusting a model? While the tradeoff in figure 5 is a good finding, I’m not sure how it translates to helping humans understand decisions made by the model.**
>
> For humans to trust in the model and understand its decisions, the explanations need to be interpretable. This has two ingredients to it.
> 1. The individual queries in the query set must have a clear interpretation to the user (since the V-IP explanations are literally a conjunction of queries from this query set along with their obtained answers). The first set of experiments (pertaining to Table 1) is aimed at evaluating the interpretability of the GPT proposed query set. This is important since Concept-QA is trained using pseudo-labels, thus if for the question ”Does it have stripes?”, Concept-QA answers ”yes” when the image has no stripes but has a checkerboard pattern (a completely different concept) then this query isn’t well-defined (from an interpretability point of view).
> 2. Assuming every query in the query set is interpretable, shorter explanations are easier to parse and understand than longer ones assuming both explanations are “sufficient” (which necessarily means that the longer explanation contains redundant information). This “sufficiency” of explanations is captured by the test accuracy. Stated precisely, if two strategies have the same classification accuracy, then we prefer the strategy that has a smaller explanation length on average (as measured by the number of queries asked by V-IP). This is based on Occam’s Razor, which is one of the most widely accepted heuristics used in science to choose between competing explanations for the same phenomena. The experiments (reported in Figure 5) are aimed at evaluating this in terms of the trade-off between test accuracy and average explanation length.
>
>
> **Similarly, when humans know that the prediction has failed – do they prefer shorter or longer explanations to debug why it failed? This study could be added.**
>
> This is an interesting avenue for future work and is out of the scope of this paper. Using interpretable machine learning algorithms for model debugging is an active area of research and it would be interesting to see how to adapt the V-IP framework to do so. However, the scope of our current work is to relieve V-IP from its query-answer annotation bottleneck and enable us to apply it successfully to many image classification datasets which was not possible before, notable among them being the ImageNet dataset.

---

> > ### Comment · Reviewer_8mpd · 2023-11-30
> > **thanks for the clarifications**
> >
> > W1. Comparisons to vision-only models: thank you for pointing this out -- I had indeed missed this table. This is useful.
> > W2: I understand the limitations to run these experiments during the rebuttal phase.  This is a minor weakness as far as I am concerned.
> > Q1. Applicability to VQA tasks:  this is a good point.  Classification datasets (most of them) are "iconic" i.e. focus on a single object.
> > W3/Q2/Q3: thanks for these insights. putting these in the paper (maybe summarized into 1-2 sentences) would help.
> >
> > I am increasing my score to 6 (marginally above threshold).

---

### Official Review · Reviewer_8xTQ · 2023-11-22

**Soundness:** 3 good
**Presentation:** 3 good
**Contribution:** 2 fair
**Rating:** 6
**Confidence:** 4

**Summary:**

This paper introduces a novel methodology for training a Concept Question-Answering system (Concept-QA) that determines the presence of a concept in an image using binary answers. This approach eliminates the need for manually annotated training data, relying instead on pseudo-labels generated by GPT and CLIP. The authors empirically validate Concept-QA on various datasets, demonstrating its accuracy in representing true concepts.

**Strengths:**

- This paper introduces a methodology for training Concept-QA that doesn't require manually annotated training data, reducing the burden of data annotation.
- Experiments on multiple datasets demonstrate the proposed method's effectiveness.
- This paper is well-written and easy to follow.

**Weaknesses:**

- It would be better to try more LLMs to show the generality of this idea.
- Limitations could be discussed. Is there anything about **the proposed method itself** that fails in certain scenarios or falls short compared to prior work?
- While the paper mentions the use of pseudo-labels from GPT and CLIP, it could discuss the interpretability and potential biases associated with these labels. Are there instances where the pseudo-labels might lead to incorrect or biased answers?

**Questions:**

While the paper highlights the advantage of not requiring manually annotated training data, it could delve deeper into the data efficiency aspect. Does Concept-QA require a large amount of unlabeled data to perform well, and how does its data efficiency compare to alternative approaches?

---

> ### Author Response · Authors · 2023-11-25
> **Response to Reviewer 8xTQ (1/2)**
>
> Thank you for your time and feedback. We would respond to each of your comments (highlighted in bold and paraphrased) below.
>
> **It would be better to try more LLMs to show the generality of this idea**
>
> Thank you for this suggestion. We used GPT in this work due to it being the state-of-the-art LLM currently, in particular their unprecedented ability to act as implicit knowledge bases  (Brown et al., 2020) (due to being trained on massive swaths of textual data from the web). Moreover, our work builds on a string of prior work (Yang et al., 2022; Menon \& Vondrick, 2022; Oikarinen et al.,2023) on concept bottleneck models where GPT has been successfully employed to provide a set of concepts for image classification problems.
>
> Having said so, we would be happy to add a few more LLMs to compare performance with GPT in the final version of the paper. It is difficult to get those results now given the time constraints of the rebuttal.
>
> **Limitations could be discussed. Is there anything about the proposed method itself that fails in certain scenarios or falls short compared to prior work?**
>
> Thank you for this suggestion. We have added a Limitations section in the Appendix of the updated paper (Section H) and referenced it at the end of the main paper. Following are our limitations.
>
> 1. Prior work on V-IP required the user to annotate datasets with query-answers which could be used to supervise the training of classifiers to provide answers at inference time. In this work, to overcome this data annotation bottleneck we propose to use GPT + CLIP to provide psuedo-labels to train our answering mechanism (the Concept-QA model). This inevitably leads to a loss in accuracy since we rely on the zero-shot performance of these large language models and vision language models. Future work would aim to address this limitation by exploring a hybrid approach where we manually annotate a small portion of the data and use it in tandem with the pseudo-labels to train the Concept-QA model.
>
> 2. The current approach is limited to the problem of image classification due to the assumptions involved in deriving our pseudo-labels, that is, the class object is the focal point of the image (mentioned in Section 3.2.3). Future work would be aimed at exploring other ways of generating pseudo-labels to extend the application of V-IP to more general vision tasks such as image captioning, scene segmentation and visual reasoning to name a few.
>
> **While the paper mentions the use of pseudo-labels from GPT and CLIP, it could discuss the interpretability and potential biases associated with these labels. Are there instances where the pseudo-labels might lead to incorrect or biased answers?**
>
> Regarding the interpretability of the labels, in the original submission, we annotated a randomly selected subset of image-query pairs (with the corresponding answers) from each of the 5 image datasets used in this paper, and evaluated the accuracy and F1-score of our trained Concept-QA model on these annotations. Higher accuracy and F1-score values indicate higher interpretability of the Concept-QA model (since it implies our model answers more correctly). Our results (Table 1) show that Concept-QA is better than the CLIP baseline and competitive with state-of-the-art VQA systems on these metrics. Moreover, following the reviewer's suggestion we have also evaluated our pseudo-labels on the same annotated dataset. Following are the results, which show a close agreement between the correctness of our pseudo-labels and Concept-QA (which was trained using these pseudo-labels).
>
> | Model         | Imagenet      | Places365     | CUB-200       | CIFAR-10      | CIFAR-100    |
> |---------------|---------------|---------------|---------------|---------------|--------------|
> |               | Acc \| F1     | Acc \| F1     | Acc \| F1     | Acc \| F1     | Acc \| F1    |
> | Concept-QA    | 0.87 \| 0.56  | 0.83 \| 0.45  | 0.80 \| 0.54  | 0.80 \| 0.62  | 0.80 \| 0.38 |
> | Pseudo-labels | 0.86 \| 0.55  | 0.82 \| 0.42  | 0.79 \| 0.53  | 0.81 \| 0.62  | 0.79 \| 0.37 |
>
> These results for the pseudo-labels have been reported in the updated paper (Appendix Section F).
>
> Finally, regarding biases in our pseudo-labels, we believe that since they are constructed from GPT and CLIP they will inherit the biases of these models. A principled way of uncovering biases from these large language models and vision language models is an active area of research and out of the scope of this current work, which is mainly focused on extending the application of V-IP (an interpretable-by-design method) to tasks where query-answer annotated data is not available.

---

> ### Author Response · Authors · 2023-11-25
> **Response to Reviewer 8xTQ (2/2)**
>
> **While the paper highlights the advantage of not requiring manually annotated training data, it could delve deeper into the data efficiency aspect. Does Concept-QA require a large amount of unlabeled data to perform well, and how does its data efficiency compare to alternative approaches?**
>
> To train our Concept-QA network, for every dataset, we annotated the entire training set using GPT + CLIP with query answer labels (recall these datasets only come with class label annotations). These query answer labels (which we call pseudo-labels since they are provided by LLMs and VLMs) are then used to train Concept-QA. Thus, Concept-QA does not need any more data than training/fine-tuning a CNN or vision transformer to perform supervised classification on these datasets.

---

### Author Response · Authors · 2023-12-02
**Summary of our Rebuttal**

We thank all the reviewers for their insightful comments that helped us improve our paper. We are glad the reviewers found our paper well-written and easy to follow. We summarize the main critiques here. Afterwards, we address each reviewer's comments individually.

1. Reviewer 8xTQ wanted a discussion on the limitations of our method which we have done in the updated paper (section H of the appendix).
2. Reviewer 8mpd and Reviewer 3dNV had some concerns over the connection between interpretability, length of query chains and test accuracy of predictions. We believe we have adequately clarified this connection in our detailed responses below. Moreover, we have added aspects of this discussion in the opening of section 5 (experiments) to better motivate our experiment design.
3. Reviewer ZeKa wanted us to reiterate the key scientific insights and contributions of our paper in the context of prior work. We have articulated this in our detailed response below.
4. Reviewer 3dNV requested more baselines to compare our proposed Concept-QA + V-IP method which we have done and added in the Experiments section of the paper and section F in the appendix.

---

### Meta-Review · Area_Chair_Qwir · 2023-12-14

**Metareview:**

Concept-QA eliminates the need for manually annotated training data, overcoming a significant bottleneck in V-IP. The paper demonstrates how Concept-QA leads to shorter and more human-understandable query chains compared to baseline methods. Concept-QA achieves competitive performance on multiple datasets, highlighting its potential for real-world applications. In general, I like the way this draft is written, easy to follow.

As these being stated, the paper primarily focuses on comparing Concept-QA to a CLIP baseline. Including additional benchmarks, such as other pseudo-label generation methods and vision-only models, would provide a more comprehensive understanding of its strengths and limitations. I also agree the current approach to measuring interpretability (query chain length) might be simplistic. Exploring alternative metrics or qualitative analyses could provide a more nuanced understanding of how Concept-QA improves interpretability. Lastly, While the paper suggests that improved QA performance leads to better V-IP, it lacks concrete evidence to support this claim. Experiments demonstrating the impact of Concept-QA on the overall V-IP system would strengthen the paper's argument.

**Justification For Why Not Higher Score:**

While the draft presents an interesting and novel idea, it could benefit from improving the following as pointed out by the reviewers: 1) Perform a thorough evaluation of the generated pseudo-labels, including their quality and potential biases. 2) Explore alternative metrics or qualitative analyses to provide a more nuanced understanding of interpretability improvements. 3) Conduct experiments to demonstrate the impact of Concept-QA on the overall V-IP system performance.

**Justification For Why Not Lower Score:**

With the clear written draft and the novelties of reducing data annotations, improving interpretability and effectiveness, the draft would be of interest to ICLR community and the method and the dataset could contribute good amount to the sub-field.

---

### Decision · Program_Chairs · 2024-01-16

Accept (poster)